# Distributed Differential Privacy in Multi-Armed Bandits

**Sayak Ray Chowdhury** [*]
Microsoft Research
Bengaluru, Karnataka, India
`t-sayakr@microsoft.com`

**Xingyu Zhou** [*]
Department of Electrical and Computer Engineering
Wayne State University
Detroit, USA
`xingyu.zhou@wayne.edu`

## Abstract

We consider the standard $K$-armed bandit problem under a distributed trust model of differential privacy (DP), which enables to guarantee privacy without a trustworthy server. Under this trust model, previous work on private bandits largely focus on achieving privacy using a shuffle protocol, where a batch of users data are randomly permuted before sending to a central server. This protocol achieves $(\varepsilon, \delta)$ or approximate-DP guarantee by sacrificing an additive $O\left(\frac{K \log T \sqrt{\log(1/\delta)}}{\varepsilon}\right)$ factor in $T$-step cumulative regret. In contrast, the optimal privacy cost to achieve a stronger $(\varepsilon, 0)$ or pure-DP guarantee under the widely used central trust model is only $\Theta\left(\frac{K \log T}{\varepsilon}\right)$, where, however, a trusted server is required. In this work, we aim to obtain a pure-DP guarantee under distributed trust model while sacrificing no more regret than that under the central trust model. We achieve this by designing a generic bandit algorithm based on successive arm elimination, where privacy is guaranteed by corrupting rewards with an equivalent discrete Laplace noise ensured by a secure computation protocol. We also show that our algorithm, when instantiated with Skellam noise and the secure protocol, ensures *Rényi differential privacy* – a stronger notion than approximate DP – under distributed trust model with a privacy cost of $O\left(\frac{K \sqrt{\log T}}{\varepsilon}\right)$. Our theoretical findings are corroborated by numerical evaluations on both synthetic and real-world data.

## 1 Introduction

The multi-armed bandit (MAB) problem provides a simple but powerful framework for sequential decision-making under uncertainty with bandit feedback, which has attracted a wide range of practical applications such as online advertising (Abe et al., 2003), product recommendations (Li et al., 2010), clinical trials (Tewari & Murphy, 2017), to name a few. Along with its broad applicability, however, there is an increasing concern of privacy risk in MAB due to its intrinsic dependence on users' feedback, which could leak users' sensitive information (Pan et al., 2019).

To alleviate the above concern, the notion of *differential privacy*, introduced by Dwork et al. (2006) in the field of computer science theory, has recently been adopted to design privacy-preserving bandit algorithms (see, e.g., Mishra & Thakurta (2015); Tossou & Dimitrakakis (2016); Shariff & Sheffet (2018)). Differential privacy (DP) provides a principled way to mathematically prove privacy guarantees against adversaries with arbitrary auxiliary information about users. To achieve this, a differentially private bandit algorithm typically relies on a well-tuned random noise to obscure each user's contribution to the output, depending on privacy levels $\varepsilon, \delta$ – smaller values lead to stronger protection but also suffer worse utility (i.e., regret). For example, the central server of a recommendation system can use random noise to perturb its statistics on each item after receiving feedback (i.e., clicks/ratings) from users. This is often termed as *central model* (Dwork et al., 2014), since the central server has the trust of its users and hence has a direct access to their raw

---

[*]Equal contributions

Table 1: Best-known performance of private MAB under different privacy models ($K$ = number of arms, $T$ = time horizon, $\Delta_a$ = reward gap of arm $a$ w.r.t. best arm, $\varepsilon, \delta, \alpha$ = privacy parameters)

| Trust Model | Privacy Guarantee | Best-Known Regret Bounds |
|---|---|---|
| Central | $(\varepsilon, 0)$-DP | $\Theta\left(\sum_{a \in [K]: \Delta_a > 0} \frac{\log T}{\Delta_a} + \frac{K \log T}{\varepsilon}\right)$ (Sajed & Sheffet, 2019) |
| Local | $(\varepsilon, 0)$-DP | $\Theta\left(\frac{1}{\varepsilon^2} \sum_{a \in [K]: \Delta_a > 0} \frac{\log T}{\Delta_a}\right)$ (Ren et al., 2020) |
| Distributed | $(\varepsilon, \delta)$-DP | $O\left(\sum_{a: \Delta_a > 0} \frac{\log T}{\Delta_a} + \frac{K \log T \sqrt{\log \frac{1}{\delta}}}{\varepsilon}\right)$ (Tenenbaum et al., 2021) |
| Distributed | $(\varepsilon, 0)$-DP | $\Theta\left(\sum_{a \in [K]: \Delta_a > 0} \frac{\log T}{\Delta_a} + \frac{K \log T}{\varepsilon}\right)$ (Theorem 1) |
| Distributed | $O(\alpha, \frac{\alpha \varepsilon^2}{2})$-RDP | $O\left(\sum_{a \in [K]: \Delta_a > 0} \frac{\log T}{\Delta_a} + \frac{K \sqrt{\log T}}{\varepsilon}\right)$ (Theorem 2) |

data. Under this model, an optimal private MAB algorithm with a pure DP guarantee (i.e., when $\delta = 0$) is proposed in Sajed & Sheffet (2019), which only incurs an *additive* $O\left(\frac{K \log T}{\varepsilon}\right)$ term in the cumulative regret compared to the standard setting when privacy is not sought after (Auer, 2002). However, this high trust model is not always feasible in practice since users may not be willing to share their raw data directly to the server. This motivates to employ a *local model* (Kasiviswanathan et al., 2011) of trust, where DP is achieved without a trusted server as each user perturbs her data prior to sharing with the server. This ensures a stronger privacy protection, but leads to a high cost in utility due to large aggregated noise from all users. As shown in Ren et al. (2020), under the local model, private MAB algorithms have to incur a *multiplicative* $1/\varepsilon^2$ factor in the regret rather than the additive one in the central model.

In attempts to recover the same utility of central model while without a trustworthy server like the local model, an intermediate DP trust model called *distributed model* has gained an increasing interest, especially in the context of (federated) supervised learning (Kairouz et al., 2021b; Agarwal et al., 2021; Kairouz et al., 2021a; Girgis et al., 2021; Lowy & Razaviyayn, 2021). Under this model, each user first perturbs her data via a local randomizer, and then sends the randomized data to a *secure* computation function. This secure function can be leveraged to guarantee privacy through aggregated noise from distributed users. There are two popular secure computation functions: *secure aggregation* (Bonawitz et al., 2017) and *secure shuffling* (Bittau et al., 2017). The former often relies on cryptographic primitives to securely aggregate users' data so that the central server only learns the aggregated result, while the latter securely shuffle users' messages to hide their source. To the best of our knowledge, distributed DP model is far less studied in online learning as compared to supervised learning, with only known results for standard $K$-armed bandits in Tenenbaum et al. (2021), where secure shuffling is adopted. Despite being pioneer work, the results obtained in this paper have several limitations: (i) The privacy guarantee is obtained only for approximate DP ($\delta > 0$) – a stronger pure DP ($\delta = 0$) guarantee is not achieved; (ii) The cost of privacy is a multiplicative $\sqrt{\log(1/\delta)}$ factor away from that of central model, leading to a higher regret bound; (iii) The secure protocol works only for binary rewards (or communication intensive for real rewards).[1]

**Our contributions.** In this work, we design the *first* communication-efficient MAB algorithm that satisfies pure DP in the distributed model while attaining the same regret bound as in the central model (see Table 1). We overcome several key challenges that arise in the design and analysis of distributed DP algorithms for bandits. We now list the challenges and our proposed solutions below.

**(a) Private and communication efficient algorithm design.** Secure aggregation (SecAgg) works only in the integer domain due to an inherent modular operation (Bonawitz et al., 2017). Hence, leveraging this in bandits to achieve distributed DP with *real* rewards needs adopting data quantization, *discrete privacy noise* and modular summation arithmetic in the algorithm design. To this end, we take a batch version of the successive arm elimination technique as a building block of our algorithm, and on top of it, employ a privacy protocol tailored to discrete privacy noise and modular operation (see Algorithm 1). Instantiating the protocol at each user with *Pólya* random noise, we

---

[1] For more general linear bandits under distributed DP via shuffling, see Chowdhury & Zhou (2022b); Garcelon et al. (2022), which also have similar limitations.

ensure that our algorithm satisfies pure DP in the distributed model. Moreover, the communication bits per-user scale only logarithmicaly with the number of participating users in each batch.

**(b) Regret analysis under pure DP with SecAgg.** While our pure DP guarantee exploits known results for discrete Laplace mechanism, the utility analysis gets challenging due to modular clipping of SecAgg. In fact, in supervised learning, no known convergence rate exists for SGD under pure DP with SecAgg (although the same is well-known under central model). This is because modular clipping makes gradient estimates biased, and hence, standard convergence guarantees using unbiased estimates do not hold. In bandits, however, we work with zeroth order observations to build estimates of arms' rewards, and require high-confidence tight tail bounds for the estimates to analyse convergence. To this end, relying on tail properties of discrete Laplace and a careful analysis of modular operation, we prove a sublinear regret rate of our algorithm, which matches the optimal one in the central model, and thus, achieves the optimal rate under pure DP (see Theorem 1).

**(c) Improved regret bound under RDP.** While our main focus were to design the first bandit algorithm with pure distributed DP that achieves the same regret rate under central model, our template protocol is general enough to obtain different privacy guarantees by tuning the noise at each user. We demonstrate this by achieving *Rényi differential privacy* (RDP) (Mironov, 2017) using a Skellam random noise. RDP is a weaker notion of privacy compared to pure DP, but it is still stronger than approximate DP. It also provides a tighter privacy accounting for composition compared to approximate DP. This is particularly useful for bandit algorithms, when users may participate in multiple rounds, necessitating the need for privacy composition. Hence, we focus on RDP with SecAgg and and show that a tighter regret bound compared to pure DP can be achieved (see Theorem 2) by proving novel tail-bound for Skellam distribution. We support our theoretical findings with extensive numerical evaluation over bandit instances generated from both synthetic and real-life data.

Finally, our analysis technique is also general enough to recover best-known regrets under central and local DP models while only using *discrete* privacy noise (see Appendix H). This is important in practice since continuous Laplace noise might leak privacy on finite computers due to floating point arithmetic (Mironov, 2012), which is a drawback of existing central and local DP MAB algorithms.

## 2 PRELIMINARIES

In this section, we formally introduce the distributed differential privacy model in bandits. Before that we recall the learning paradigm in multi-armed bandits and basic differential privacy definitions.

**Learning model and regret in MAB.** At each time slot $t \in [T] := \{1, \ldots, T\}$, the agent (e.g., recommender system) selects an arm $a \in [K]$ (e.g., an advertisement), recommends it to a new user $t$ and obtains an i.i.d reward $r_t$ (e.g., a rating indicating how much she likes it), which is sampled from a distribution over $[0, 1]$ with mean given by $\mu_a$. Let $a^* := \operatorname{argmax}_{a \in [K]} \mu_a$ be the arm with the highest mean and denote $\mu^* := \mu_{a^*}$ for simplicity. Let $\Delta_a := \mu^* - \mu_a$ be the gap of the expected reward between the optimal arm $a^*$ and any other arm $a$. Further, let $N_a(t)$ be the total number of times that arm $a$ has been recommended to first $t$ users. The goal of the agent is to maximize its total reward, or equivalently to minimize the cumulative expected pseudo-regret, defined as

$$\mathbb{E}\left[\operatorname{Reg}(T)\right] := T \cdot \mu^* - \mathbb{E}\left[\sum\nolimits_{t=1}^{T} r_t\right] = \mathbb{E}\left[\sum\nolimits_{a \in [K]} \Delta_a N_a(T)\right].$$

**Differential privacy.** Let $\mathcal{D} = [0, 1]$ be the data universe, and $n \in \mathbb{N}$ be the number of *unique* users. we say $D, D' \in \mathcal{D}^n$ are neighboring datasets if they only differ in one user's reward preference for some $i \in [n]$. We have the following standard definition of differential privacy (Dwork et al., 2006).

**Definition 1** (Differential Privacy). *For $\varepsilon, \delta > 0$, a randomized mechanism $\mathcal{M}$ satisfies $(\varepsilon, \delta)$-DP if for all neighboring datasets $D, D'$ and all events $\mathcal{E}$ in the range of $\mathcal{M}$, we have*

$$\mathbb{P}\left[\mathcal{M}(D) \in \mathcal{E}\right] \le e^{\varepsilon} \cdot \mathbb{P}\left[\mathcal{M}(D') \in \mathcal{E}\right] + \delta.$$

The special case of $(\varepsilon, 0)$-DP is often referred to as *pure differential privacy*, whereas, for $\delta > 0$, $(\varepsilon, \delta)$-DP is referred to as *approximate differential privacy*. We also consider a related notion of privacy called Rényi differential privacy (RDP) Mironov (2017), which allows for a tighter composition compared to approximate differential privacy.

**Definition 2** (Rényi Differential Privacy). *For $\alpha > 1$, a randomized mechanism $\mathcal{M}$ satisfies $(\alpha, \varepsilon(\alpha))$-RDP if for all neighboring datasets $D, D'$, we have $D_\alpha(\mathcal{M}(D), \mathcal{M}(D')) \le \varepsilon(\alpha)$, where*

$D_\alpha(P, Q)$ *is the Rényi divergence (of order $\alpha$) of the distribution $P$ from the distribution $Q$, and is given by* $D_\alpha(P, Q) := \frac{1}{\alpha-1} \log \left( \mathbb{E}_{x \sim Q} \left[ \left( \frac{P(x)}{Q(x)} \right)^\alpha \right] \right).$

**Distributed differential privacy.** A distributed bandit learning protocol $\mathcal{P} = (\mathcal{R}, \mathcal{S}, \mathcal{A})$ consists of three parts: (i) a (local) randomizer $\mathcal{R}$ at each user's side, (ii) an intermediate secure protocol $\mathcal{S}$, and (iii) an analyzer $\mathcal{A}$ at the central server. Each user $i$ first locally apply the randomizer $\mathcal{R}$ on its raw data (i.e., reward) $D_i$, and sends the randomized data to a secure computation protocol $\mathcal{S}$ (e.g., secure aggregation or shuffling). This intermediate secure protocol $\mathcal{S}$ takes a batch of users' randomized data and generates inputs to the central server, which utilizes an analyzer $\mathcal{A}$ to compute the output (e.g., action) using received messages from $\mathcal{S}$.

The secure computation protocol $\mathcal{S}$ has two main variations: *secure shuffling* and *secure aggregation*. Both of them essentially work with a batch of users' randomized data and guarantee that the central server cannot infer any individual's data while the total noise in the inputs to the analyzer provides a high privacy level. To adapt both into our MAB protocol, it is natural to divide participating users into batches. For each batch $b \in [B]$ with $n_b$ users, the outputs of $\mathcal{S}$ is given by $\mathcal{S} \circ \mathcal{R}^{n_b}(D) := \mathcal{S}(\mathcal{R}(D_1), \dots, \mathcal{R}(D_{n_b}))$. The goal is to guarantee that the the view of all $B$ batches' outputs satisfy DP. To this end, we define a (composite) mechanism

$$\mathcal{M}_\mathcal{P} = (\mathcal{S} \circ \mathcal{R}^{n_1}, \dots, \mathcal{S} \circ \mathcal{R}^{n_B}),$$

where each individual mechanism $\mathcal{S} \circ \mathcal{R}^{n_b}$ operates on $n_b$ users' rewards, i.e., on a dataset from $\mathcal{D}^{n_b}$. With this notation, we have the following definition of distributed differential privacy.

**Definition 3** (Distributed DP). *A protocol $\mathcal{P} = (\mathcal{R}, \mathcal{S}, \mathcal{A})$ is said to satisfy DP (or RDP) in the distributed model if the mechanism $\mathcal{M}_\mathcal{P}$ satisfies Definition 1 (or Definition 2).*

In the central DP model, the privacy burden lies with a central server (in particular, analyzer $\mathcal{A}$), which needs to inject necessary random noise to achieve privacy. On the other hand, in the local DP model, each user's data is privatized by local randomizer $\mathcal{R}$. In contrast, in the distributed DP model, privacy without a trusted central server is achieved by ensuring that the inputs to the analyzer $\mathcal{A}$ already satisfy differential privacy. Specifically, by properly designing the intermediate protocol $\mathcal{S}$ and the noise level in the randomizer $\mathcal{R}$, one can ensure that the final added noise in the aggregated data over a batch of users matches the noise that would have otherwise been added in the central model by the trusted server. Through this, distributed DP model provides the possibility to achieve the same level of utility as the central model while without a trustworthy central server.

## 3 A GENERIC ALGORITHM FOR PRIVATE BANDITS

In this section, we propose a generic algorithmic framework (Algorithm 1) for multi-armed bandits under the distributed privacy model.

**Batch-based successive arm elimination.** Our algorithm builds upon the classic idea of successive arm elimination (Even-Dar et al., 2006) with the additional incorporation of batches and a black-box protocol $\mathcal{P} = (\mathcal{R}, \mathcal{S}, \mathcal{A})$ to achieve distributed differential privacy. It divides the time horizon $T$ into batches of exponentially increasing size and eliminates sub-optimal arms successively. To this end, for each active arm $a$ at batch $b$, it first prescribes arm $a$ to a *batch* of $l(b) = 2^b$ *new* users. After pulling the prescribed action $a$, each user applies the local randomizer $\mathcal{R}$ to her reward and sends the randomized reward to the intermediary function $\mathcal{S}$, which runs a secure computation protocol (e.g., secure aggregation or secure shuffling) over the total $l(b)$ number of randomized rewards. Then, upon receiving the outputs of $\mathcal{S}$, the server applies the analyzer $\mathcal{A}$ to compute the the sum of rewards for batch $b$ when pulling arm $a$ (i.e., $R_a(b)$), which in turn gives the new mean estimate $\widehat{\mu}_a(b)$ of arm $a$ after being divided by the total pulls $l(b)$. Then, upper and lower confidence bounds, $\text{UCB}_a(b)$ and $\text{LCB}_a(b)$, respectively, are computed around the mean estimate $\widehat{\mu}_a(b)$ with a properly chosen confidence width $\beta(b)$. Finally, after the iteration over all active arms in batch $b$ (denoted by the set $\Phi(b)$), it adopts the standard arm elimination criterion to remove all obviously sub-optimal arms, i.e., it removes an arm $a$ from $\Phi(b)$ if $\text{UCB}_a(b)$ falls below $\text{LCB}_{a'}(b)$ of any other arm $a' \in \Phi(b)$. It now only remains to design a distributed DP protocol $\mathcal{P}$.

There is one key difference between our algorithm and the VB-SDP-AE algorithm in Tenenbaum et al. (2021). At the start of a batch, VB-SDP-AE uses all the past data to compute reward estimates. In contrast, we adopt the idea of *forgetting* and use only the data of the last completed batch.

---

**Algorithm 1** Private Batch-Based Successive Arm Elimination

---

1: **Parameters:** # arms $K$, Time horizon $T$, privacy level $\varepsilon > 0$, Confidence radii $\{\beta(b)\}_{b \geq 1}$
2: **Initialize:** Batch count $b = 1$, Active arm set $\Phi(b) = \{1, \ldots, K\}$, Estimate $\widehat{\mu}_a(1) = 0$, $\forall a \in [K]$
3: **for** batch $b = 1, 2, \ldots$ **do**
4:     Set batch size $l(b) = 2^b$
5:     **for** each active arm $a \in \Phi(b)$ **do**
6:         **for** each new user $i$ from 1 to $l(b)$ **do**
7:             Pull arm $a$ and generate reward $r_a^i(b)$
8:             Send randomized data $y_a^i(b) = \mathcal{R}(r_a^i(b))$ to $\mathcal{S}$           // randomizer
9:             If total number of pulls reaches $T$, **exit**
10:         **end for**
11:         Send messages $\widehat{y}_a(b) = \mathcal{S}(\{y_a^i(b)\}_{1 \leq i \leq l(b)})$ to $\mathcal{A}$     // secure computation
12:         Compute the sum of rewards $R_a(b) = \mathcal{A}(\widehat{y}_a(b))$       // analyzer
13:         Compute mean estimate $\widehat{\mu}_a(b) = R_a(b)/l(b)$
14:         Compute confidence bounds $\text{UCB}_a(b) = \widehat{\mu}_a(b) + \beta(b)$ and $\text{LCB}_a(b) = \widehat{\mu}_a(b) - \beta(b)$
15:     **end for**
16:     Update active set of arms: $\Phi(b+1) = \left\{ a \in \Phi(b) : \text{UCB}_a(b) \geq \max_{a' \in \Phi(b)} \text{LCB}_{a'}(b) \right\}$
17: **end for**
18: Subroutine: Local Randomizer $\mathcal{R}$ (**Input:** $x_i \in [0,1]$, **Output:** $y_i$)
19: **Require:** precision $g \in \mathbb{N}$, modulo $m \in \mathbb{N}$, batch size $n \in \mathbb{N}$, privacy level $\varepsilon$
20: Encode $x_i$ as $\widehat{x}_i = \lfloor x_i g \rfloor + \text{Ber}(x_i g - \lfloor x_i g \rfloor)$
21: Generate discrete noise $\eta_i$ (depending on $n, \varepsilon, g$)       // random noise generator
22: Add noise and modulo clip $y_i = (\widehat{x}_i + \eta_i) \bmod m$
23: Subroutine: Secure Aggregation $\mathcal{S}$ (**Input:** $y_1, \ldots, y_n$, **Output:** $\widehat{y}$)
24: **Require:** modulo $m \in \mathbb{N}$
25: Securely compute $\widehat{y} = (\sum_{i=1}^n y_i) \bmod m$       // black-box function
26: Subroutine: Analyzer $\mathcal{A}$ (**Input:** $\widehat{y}$, **Output:** $z$)
27: **Require:** precision $g \in \mathbb{N}$, modulo $m \in \mathbb{N}$, batch size $n \in \mathbb{N}$, accuracy level $\tau \in \mathbb{R}$
28: **if** $\widehat{y} > ng + \tau$ **then**
29:     set $z = (\widehat{y} - m)/g$       // correction for underflow
30: **else** set $z = \widehat{y}/g$

---

**Distributed DP protocol via discrete privacy noise.** Inspired by Balle et al. (2020); Cheu & Yan (2021), we provide a general template protocol $\mathcal{P}$ for the distributed DP model, which relies only on *discrete privacy noise*.

Local randomizer $\mathcal{R}$ receives each user $i$'s real-valued data $x_i$ and encodes it as an integer via fixed-point encoding with precision $g > 0$ and randomized rounding. Then, it generates a discrete noise, which depends on the specific privacy-regret trade-off requirement (to be discussed later under specific mechanisms). Next, it adds the random noise to the encoded reward, clips the sum with modulo $m \in \mathbb{N}$ and sends the final integer $y_i$ as input to secure computation function $\mathcal{S}$.

We mainly focus on secure aggregation (SecAgg) for $\mathcal{S}$ here.[2] SecAgg is treated as a black-box function as in previous work on supervised learning (Kairouz et al., 2021a), which implements the following procedure: given $n$ users and their randomized messages $y_i \in \mathbb{Z}_m$ (i.e., integer in $\{0, 1, \ldots, m-1\}$) obtained via $\mathcal{R}$, the SecAgg function $\mathcal{S}$ *securely* computes the modular sum of the $n$ messages, $\widehat{y} = (\sum_{i=1}^n y_i) \bmod m$, while revealing no further information on individual messages to a potential attacker, ensuring that it is *perfectly secure*. Details of engineering implementations of SecAgg is beyond the scope of this paper, see Appendix G for a brief discussion on this.

The job of analyzer $\mathcal{A}$ is to compute the sum of rewards within a batch as accurately as possible. It uses an accuracy parameter $\tau \in \mathbb{R}$ and $g$ to correct for possible underflow due to modular operation and bias due to encoding. To sum it up, the end goal of our protocol $\mathcal{P} = (\mathcal{R}, \mathcal{S}, \mathcal{A})$ is to ensure that it provides the required privacy protection while guaranteeing an output $z \approx \sum_{i=1}^n x_i$ with high probability, which is the key to our privacy and regret analysis in the following sections.

---

[2]Instead of SecAgg, we can also use secure shuffling as $\mathcal{S}$ (see Appendix D.2), since each has advantages over the other. The high-level idea is same for both techniques; i.e., to ensure that after receiving messages from $\mathcal{S}$, the server cannot distinguish each individual's message.

## 4 ACHIEVING PURE DP IN THE DISTRIBUTED MODEL

In this section, we show that Algorithm 1 achieves pure-DP in the distributed DP model via secure aggregation. To do so, we need to carefully determine the amount of (discrete) noise in $\mathcal{R}$ so that the total noise in a batch provides $(\varepsilon, 0)$-DP. One natural choice is the discrete Laplace noise.

**Definition 4** (Discrete Laplace Distribution). *Let $b > 0$. A random variable $X$ has a discrete Laplace distribution with scale parameter $b$, denoted by $\mathbf{Lap}_{\mathbb{Z}}(b)$, if it has a p.m.f. given by*

$$\forall x \in \mathbb{Z}, \quad \mathbb{P}\left[X = x\right] = \frac{e^{1/b} - 1}{e^{1/b} + 1} \cdot e^{-|x|/b}.$$

A key property of discrete Laplace that we will use is its *infinite divisibility*, which allows us to simulate it in a distributed way (Goryczka & Xiong, 2015, Theorem 5.1).

**Fact 1** (Infinite Divisibility of Discrete Laplace). *A random variable $X$ has a Pólya distribution with parameters $r > 0, \beta \in [0, 1]$, denoted by $\mathbf{Pólya}(r, \beta)$, if it has a p.m.f. given by*

$$\forall x \in \mathbb{N}, \quad \mathbb{P}\left[X = x\right] = \frac{\Gamma(x + r)}{x!\Gamma(r)}\beta^x(1 - \beta)^r.$$

*Now, for any $n \in \mathbb{N}$, let $\{\gamma_i^+, \gamma_i^-\}_{i \in [n]}$ be $2n$ i.i.d samples[3] from $\mathbf{Pólya}(1/n, e^{-1/b})$, then the random variable $\sum_{i=1}^{n}(\gamma_i^+ - \gamma_i^-)$ is distributed as $\mathbf{Lap}_{\mathbb{Z}}(b)$.*

Armed with the above fact and the properties of discrete Laplace noise (see Fact 3 in Appendix K), we are able to obtain the following main theorem, which shows that the same regret as in the central model is achieved under the distributed model via SecAgg.

**Theorem 1** (Pure-DP via SecAgg). *Fix $\varepsilon > 0$ and $T \in \mathbb{N}$. For each batch $b$, let noise for the $i$-th user in the batch be $\eta_i = \gamma_i^+ - \gamma_i^-$, where $\gamma_i^+, \gamma_i^- \overset{i.i.d.}{\sim} \mathbf{Pólya}(1/n, e^{-\varepsilon/g})$, set $n = l(b), g = \lceil \varepsilon\sqrt{n} \rceil$, $\tau = \lceil \frac{g}{\varepsilon}\log(2T) \rceil$ and $m = ng + 2\tau + 1$. Then, Algorithm 1 achieves $(\varepsilon, 0)$-DP in the distributed model. Moreover, setting $\beta(b) = O\left(\sqrt{\frac{\log(|\Phi(b)|b^2 T)}{2l(b)}} + \frac{2\log(|\Phi(b)|b^2 T)}{\varepsilon l(b)}\right)$, it enjoys expected regret*

$$\mathbb{E}\left[Reg(T)\right] = O\left(\sum_{a \in [K]: \Delta_a > 0} \frac{\log T}{\Delta_a} + \frac{K \log T}{\varepsilon}\right).$$

**Theorem 1 achieves optimal regret under pure DP.** Theorem 1 achieves the same regret bound as the one achieved in Sajed & Sheffet (2019) under the central trust model with *continuous* Laplace noise. Moreover, it matches the lower bound obtained under pure DP in Shariff & Sheffet (2018), indicating the bound is indeed tight. Note that, we achieve this rate under distributed trust model – a stronger notion of privacy protection than the central model – while using only discrete noise.

**Communication bits.** Algorithm 1 needs to communicate $O(\log m)$ bits per user to the secure protocol $\mathcal{S}$, i.e., communicating bits scales logarithmically with the batch size. In contrast, the number of communication bits required in existing distributed DP bandit algorithms that work with real-valued rewards (as we consider here) scale polynomially with the batch size (Chowdhury & Zhou, 2022b; Garcelon et al., 2022).

**Remark 1** (Pure DP via Secure Shuffling). *It turns out that one can achieve same privacy and regret guarantees (orderwise) using a relaxed SecAgg protocol. Building on this result, we also establish pure DP under shuffling while again maintaining the same regret bound as the central model (see Theorem 3 in Appendix D.2). This improves the state-of-the-art result for MAB with shuffling (Tenenbaum et al., 2021) in terms of both privacy and regret.*

## 5 ACHIEVING RDP IN THE DISTRIBUTED MODEL

A natural question to ask is whether one can get a better regret performance by sacrificing a small amount of privacy. We consider the notion of RDP (see Definition 2), which is a weaker notion of privacy than pure DP. However, it avoids the possible catastrophic privacy failure in approximate DP, and also provides a tighter privacy accounting for composition (Mironov, 2017).

---

[3]One can sample from $\mathbf{Pólya}$ as follows. First, sample $\lambda \sim \mathbf{Gamma}(r, \beta/(1-\beta))$ and then use it to sample $X \sim \mathbf{Poisson}(\lambda)$, which is known to follow $\mathbf{Pólya}(r, \beta)$ distribution (Goryczka & Xiong, 2015).

To achieve RDP guarantee using discrete noise, we consider the Skellam distribution – which has recently been introduced in private federated learning (Agarwal et al., 2021). A key challenge in the regret analysis of our bandit algorithm is to characterize the tail property of Skellam distribution. This is different from federated learning, where characterizing the variance renders sufficient. In Proposition 1, we prove that Skellam has sub-exponential tails, which not only is the key to our regret analysis, but could also be of independent interest. Below is the formal definition of Skellam.

**Definition 5** (Skellam Distribution). *A random variable $X$ has a Skellam distribution with mean $\mu$ and variance $\sigma^2$, denoted by $\boldsymbol{Sk}(\mu, \sigma^2)$, if it has a probability mass function given by*

$$\forall x \in \mathbb{Z}, \quad \mathbb{P}\left[X = x\right] = e^{-\sigma^2} I_{x-\mu}(\sigma^2),$$

*where $I_\nu(\cdot)$ is the modified Bessel function of the first kind.*

To sample from Skellam distribution, one can rely on existing procedures for Poisson samples. This is because if $X = N_1 - N_2$, where $N_1, N_2 \overset{\text{i.i.d.}}{\sim} \textbf{Poisson}(\sigma^2/2)$, then $X$ is $\textbf{Sk}(0, \sigma^2)$ distributed. Moreover, due to this fact, Skellam is *closed* under summation, i.e., if $X_1 \sim \textbf{Sk}(\mu_1, \sigma_1^2)$ and $X_2 \sim \textbf{Sk}(\mu_2, \sigma_2^2)$, then $X_1 + X_2 \sim \textbf{Sk}(\mu_1 + \mu_2, \sigma_1^2 + \sigma_2^2)$.

**Proposition 1** (Sub-exponential Tail of Skellam). *Let $X \sim \boldsymbol{Sk}(0, \sigma^2)$. Then, $X$ is $(2\sigma^2, \frac{\sqrt{2}}{2})$-sub-exponential. Hence, for any $p \in (0, 1]$, with probability at least $1 - p$,*

$$|X| \le 2\sigma\sqrt{\log(2/p)} + \sqrt{2}\log(2/p).$$

With the above result, we can establish the following privacy and regret guarantee of Algorithm 1.

**Theorem 2** (RDP via SecAgg). *Fix $\varepsilon > 0$, $T \in \mathbb{N}$ and a scaling factor $s \ge 1$. For each batch $b$, let noise for the $i$-th user be $\eta_i \sim \boldsymbol{Sk}(0, \frac{g^2}{n\varepsilon^2})$, set $n = l(b), g = \lceil s\varepsilon\sqrt{n} \rceil, \tau = \lceil \frac{2g}{\varepsilon}\sqrt{\log(2T)} + \sqrt{2}\log(2T)\rceil$ and $m = ng + 2\tau + 1$. Then, Algorithm 1 achieves $(\alpha, \widehat{\varepsilon}(\alpha))$-RDP in the distributed model for all $\alpha = 2, 3, \ldots$, with $\widehat{\varepsilon}(\alpha) = \frac{\alpha\varepsilon^2}{2} + \min\left\{\frac{(2\alpha-1)\varepsilon^2}{4s^2} + \frac{3\varepsilon}{2s^3}, \frac{3\varepsilon^2}{2s}\right\}$. Moreover, setting $\beta(b) = O\left(\sqrt{\frac{\log(|\Phi(b)|b^2T)}{2l(b)}} + \frac{(1+1/s)\log(|\Phi(b)|b^2T)}{\varepsilon l(b)}\right)$, it enjoys the expected regret*

$$\mathbb{E}\left[Reg(T)\right] = O\left(\sum_{a \in [K]: \Delta_a > 0} \frac{\log T}{\Delta_a} + \frac{K\sqrt{\log T}}{\varepsilon} + \frac{K\log T}{s\varepsilon}\right).$$

**Privacy-Regret-Communication Trade-off.** Observe that the scaling factor $s$ allows us to achieve different trade-offs. If $s$ increases, both privacy and regret performances improve. In fact, for a sufficiently large value of $s$, the third term in the regret bound becomes sufficiently small, and we obtain an improved regret bound compared to Theorem 1. Moreover, the RDP privacy guarantee improves to $\widehat{\varepsilon}(\alpha) \approx \frac{\alpha\varepsilon^2}{2}$, which is the standard RDP rate for Gaussian mechanism (Mironov, 2017). However, a larger $s$ leads to an increase of communicating bits per user, but only grows logarithmically, since Algorithm 1 needs to communicate $O(\log m)$ bits to the secure protocol $\mathcal{S}$.

**RDP to Approximate DP.** To shed more insight on Theorem 2, we convert our RDP guarantee to approximate DP for a sufficiently large $s$. It holds that under the setup of Theorem 2, for sufficiently large $s$, one can achieve $(O(\varepsilon), \delta)$-DP with regret $O\left(\sum_{a:\Delta_a > 0} \frac{\log T}{\Delta_a} + \frac{K\sqrt{\log T \log(1/\delta)}}{\varepsilon}\right)$ (via Lemma 10 in Appendix K). Implication of this conversion is three-fold. **First**, this regret bound is $O(\sqrt{\log T})$ factor tighter than that achieved by Tenenbaum et al. (2021) using a shuffle protocol. **Second**, it yields a better regret performance compared to the bound achieved under $(\varepsilon, 0)$-DP in Theorem 1 when the privacy budget $\delta > 1/T$. This observation is consistent with the fact that a weaker privacy guarantee typically warrants a better utility bound. **Third**, this conversion via RDP also yields a gain of $O(\sqrt{\log(1/\delta)})$ in the regret when dealing with privacy composition (e.g., when participating users across different batches are *not unique*) as compared to Tenenbaum et al. (2021) that only relies on approximate DP (see Appendix I for details). This results from the fact that RDP provides a tighter composition compared to approximate DP.

**Remark 2** (Achieving RDP with discrete Gaussian). *One can also achieve RDP using discrete Gaussian noise (Canonne et al., 2020). Here, we work with Skellam noise since it is closed under summation and enjoys efficient sampling procedure as opposed to discrete Gaussian (Agarwal et al., 2021). Nevertheless, as a proof of flexibility of our proposed framework, we show in Appendix F that Algorithm 1 with discrete Gaussian noise can guarantee RDP with a similar regret bound.*

## 6 KEY TECHNIQUES: OVERVIEW

Now, we provide an overview of the key techniques behind our privacy and regret guarantees. We show that the results of Theorem 1 and 2 can be obtained via a clean generic analytical framework, which not only covers the analysis of distributed pure DP/RDP with SecAgg, but also offers a unified view of private MAB under central, local and distributed DP models.

As in many private learning algorithms, the key is to characterize the impact of added privacy noise on the utility. In our case, this reduces to capturing the tail behavior of total noise $n_a(b) := R_a(b) - \sum_{i=1}^{l(b)} r_a^i(b)$ added at each batch $b$ for each active arm $a$. The following lemma gives a generic regret bound of our algorithm under mild tail assumptions on $n_a(b)$.

**Lemma 1** (Generic regret). *Let there exist constants $\sigma, h > 0$ such that, with probability $\geq 1 - p$, $|n_a(b)| \leq \mathcal{N} := O\left(\sigma\sqrt{\log(KT/p)} + h\log(KT/p)\right)$ for all $b \geq 1$, $a \in [K]$. Then, setting confidence radius $\beta(b) = O\left(\sqrt{\log(KT/p)/l(b)} + \mathcal{N}/l(b)\right)$ and $p = 1/T$, Algorithm 1 enjoys expected regret*

$$\mathbb{E}[Reg(T)] = O\Big(\sum\nolimits_{a \in [K]:\Delta_a > 0} \frac{\log T}{\Delta_a} + K\sigma\sqrt{\log T} + Kh\log T\Big).$$

An acute reader may note that the bound $\mathcal{N}$ on the noise is the tail bound of sub-exponential distribution and it reduces to the bound for sub-Gaussian tail if $h = 0$. Our SecAgg protocol $\mathcal{P}$ with discrete Laplace noise (as in Theorem 1) satisfy this bound with $\sigma = \sqrt{2}/\varepsilon$, $h = 1/\varepsilon$. Similarly, our protocol with Skellam noise (as in Theorem 2) satisfy this bound with $\sigma = O(1/\varepsilon)$, $h = 1/(s\varepsilon)$. Therefore, we can build on the above general result to directly obtain our regret bounds. In the following, we present the high-level idea behind privacy and regret analysis in distributed DP model.

**Privacy.** For distributed DP, by definition, the view of the server during the entire algorithm needs to be private. Since each user only contributes once,[4] by parallel-composition of DP, it suffices to ensure that each view $\hat{y}_a(b)$ (line 11 in Algorithm 1) is private. To this end, under SecAgg, the distribution of $\hat{y}_a(b)$ can be simulated via $(\sum_i y_i) \bmod m$, which further reduces to $(\sum_i \hat{x}_i + \eta_i) \bmod m$ by the distributive property of modular sum. Now, consider a mechanism $\mathcal{M}$ that accepts an input dataset $\{\hat{x}_i\}_i$ and outputs $\sum_i(\hat{x}_i + \eta_i)$. By post-processing, it suffices to show that $\mathcal{M}$ satisfies pure DP or RDP. To this end, the variance $\sigma_{tot}^2$ of the total noise $\sum_i \eta_i$ needs to scale with the sensitivity of $\sum_i \hat{x}_i$. Thus, each user within a batch only needs to add a proper noise with variance of $\sigma_{tot}^2/n$. Finally, by the particular distribution properties of the noise, one can show that $\mathcal{M}$ is pure DP or RDP, and hence, obtain the privacy guarantees.

**Regret.** Thanks to Lemma 1, we only need to focus on the tail of $n_a(b)$. To this end, fix any batch $b$ and arm $a$. We have $\hat{y} = \hat{y}_a(b)$, $x_i = r_a^i(b)$, $n = l(b)$ for $\mathcal{P}$ in Algorithm 1 and we need to establish that with probability at least $1 - p$, for some $\sigma$ and $h$,

$$\left|\mathcal{A}(\hat{y}) - \sum\nolimits_i x_i\right| \leq O\left(\sigma\sqrt{\log(1/p)} + h\log(1/p)\right). \tag{1}$$

To get the bound, inspired by Balle et al. (2020); Cheu & Yan (2021), we divide the LHS into Term (i) $= |\mathcal{A}(\hat{y}) - \sum_i \hat{x}_i/g|$ and Term (ii) $= |\sum_i \hat{x}_i/g - \sum_i x_i|$, where Term (i) captures the error due to privacy noise and modular operation, while Term (ii) captures the error due to random rounding. In particular, Term (ii) can be easily bounded via sub-Gaussian tail since the noise is bounded. Term (i) needs care for the possible underflow due to modular operation by considering two different cases (see line 28-30 in Algorithm 1). In both cases, one can show that Term (i) is upper bounded by $\tau/g$ with high probability, where $\tau$ is the tail bound on the total privacy noise $\sum_i \eta_i$. Thus, depending on particular privacy noise and parameter choices, one can find $\sigma$ and $h$ such that equation 1 holds, and hence, obtain the corresponding regret bound by Lemma 1.

**Remark 3.** *As a by-product of our generic analysis technique, Algorithm 1 and privacy protocol $\mathcal{P}$ along with Lemma 1 provide a new and structured way to design and analyze private MAB algorithms under central and local models with discrete private noise (see Appendix H for details). This enables us to reap the benefits of working with discrete noise (e.g., finite-computer representations, bit communications) in all three trust models (central, local and distributed).*

---

[4]This assumption that users only contributes once is adopted in nearly all previous works for privacy analysis in bandits. We also provide a privacy analysis for returning users via RDP, see Appendix I.

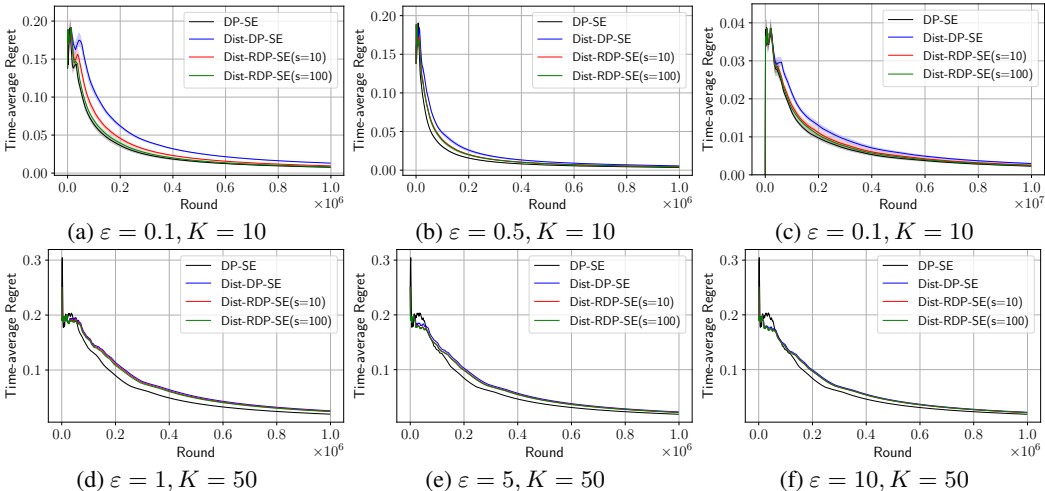

Figure 1: Comparison of time-average regret for Dist-DP-SE, Dist-RDP-SE, and DP-SE. **Top:** Synthetic Gaussian bandit instances with (a, b) large reward gap (easy instance) and (c) small reward gap (hard instance). **Bottom:** Bandit instances generated from MSLR-WEB10K learning to rank dataset.

## 7 SIMULATION RESULTS

We empirically evaluate the regret performance of our successive elimination scheme with SecAgg protocol (Algorithm 1) under distributed trust model, which we abbreviate as Dist-DP-SE and Dist-RDP-SE when the randomizer $\mathcal{R}$ is instantiated with Pólya noise (for pure DP) and Skellam noise (for RDP), respectively. We compare them with the DP-SE algorithm of Sajed & Sheffet (2019) that achieves optimal regret under pure DP in the central model, but works only with *continuous* Laplace noise. We fix confidence level $p = 0.1$ and study comparative performances under varying privacy levels ($\varepsilon < 1$ for synthetic data, $\varepsilon \geq 1$ for real data). We plot time-average regret $\text{Reg}(T)/T$ in Figure 1 by averaging results over 20 randomly generated bandit instances.

**Bandit instances.** In the *top panel*, similar to Vaswani et al. (2020), we consider easy and hard MAB instances with $K = 10$ arms: in the former, arm means are sampled uniformly in $[0.25, 0.75]$, while in the latter, those are sampled in $[0.45, 0.55]$. We consider real rewards – sampled from Gaussian distribution with aforementioned means and projected to $[0, 1]$. In the *bottom panel*, we generate bandit instances from Microsoft Learning to Rank dataset MSLR-WEB10K (Qin & Liu, 2013). The dataset consists of 1,200,192 rows and 138 columns, where each row corresponds to a query-url pair. The first column is relevance label 0, 1, ... , 4 of the pair, which we take as rewards. The second column denotes the query id, and the rest 136 columns denote contexts of a query-url pair. We cluster the data by running K-means algorithm with $K = 50$. We treat each cluster as a bandit arm with mean reward as the empirical mean of the individual ratings in the cluster. This way, we obtain a bandit setting with number of arms $K = 50$.

**Observations.** We observe that as $T$ becomes large, the regret performance of Dist-DP-SE matches the regret of DP-SE. The slight gap in small $T$ regime is the cost that we pay to achieve distributed privacy using discrete noise without access to a trusted server (for higher $\varepsilon$ value, this gap is even smaller). In addition, we find that a relatively small scaling factor ($s = 10$) provides a considerable gain in regret under RDP compared to pure DP, especially when $\varepsilon$ is small (i.e., when the cost of privacy is not dominated by the non-private part of regret). The experimental findings are consistent with our theoretical results. Here, we note that our simulations are proof-of-concept only and we did not tune any hyperparameters. More details and additional plots are given in Appendix J.[5]

**Concluding remarks.** We show that MAB under distributed trust model can achieve pure DP while maintaining the same regret under central model. In addition, RDP is also achieved in MAB under distributed trust model for the first time. Both results are obtained via a unified algorithm design and performance analysis. More importantly, our work also opens the door to a promising and interesting research direction – private online learning with distributed DP guarantees, including contextual bandits and reinforcement learning.

---

[5]Code is available at https://github.com/sayakrc/Differentially-Private-Bandits.

## 8   ACKNOWLEDGEMENTS

XZ is supported in part by NSF CNS-2153220. XZ would like to thank Albert Cheu for insightful discussion on achieving pure DP via shuffling.

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

## A    OTHER RELATED WORK

**Private Multi-Armed Bandits.**  In addition to stochastic multi-armed bandits under the central model in Mishra & Thakurta (2015); Tossou & Dimitrakakis (2016); Sajed & Sheffet (2019), different variants of differentially private bandits have been studied, including adversarial bandits (Tossou & Dimitrakakis, 2017; Agarwal & Singh, 2017), heavy-tailed bandits (Tao et al., 2022), combinatorial semi-bandits (Chen et al., 2020), and cascading bandits (Wang et al., 2022). MAB under the local model is first studied in Ren et al. (2020) for pure DP and in Zheng et al. (2020) for appromixate DP. The local model have also been considered in Tao et al. (2022); Chen et al. (2020); Wang et al. (2022); Zhou & Tan (2021). Motivated by the regret gap between the central model and local model (see Table 1), Tenenbaum et al. (2021) consider MAB in the distributed model via secure shuffling where, however, only approximate DP is achieved and the resultant regret bound still has a gap with respect to the one under the central model.

**Private Contextual Bandits.**  In contextual bandits, in addition to the reward, the contexts are also sensitive information that need to be protected. However, a straightforward adaptation of the standard central DP in contextual bandits will lead to a linear regret, as proved in Shariff & Sheffet (2018). Thus, to provide a meaningful regret under the central model for contextual bandits, a relaxed version of DP called *joint differential privacy* is considered, which rougly means that the change of any user would not change the actions prescribed to all other users, but it allows the change of the action prescribed to herself. Under this central JDP, Shariff & Sheffet (2018) establishes a regret bound of $\widetilde{O}(\sqrt{T})$ for a private variant of LinUCB. On the other hand, contextual linear bandits under the local model incurs a regret bound of $\widetilde{O}(T^{3/4})$ (Zheng et al., 2020). Very recently, motivated by the regret gap between the central and local model, two concurrent works consider contextual linear bandits in the distributed model (via secure shuffling only) (Chowdhury & Zhou, 2022b; Garcelon et al., 2022). In particular, Chowdhury & Zhou (2022b) show that a $\widetilde{O}(T^{3/5})$ regret bound is achieved under the distributed model via secure shuffling.

**Private Reinforcement Learning (RL).** RL is a generalization of contextual bandits in that contextual bandits can be viewed as a finite-horizon RL with horizon $H = 1$. This not only directly means that one also has to consider JDP in the central model for RL, but implies that RL becomes harder for privacy protection due to the additional state transition (i.e., $H > 1$). Tabular episodic RL under central JDP is first studied in Vietri et al. (2020) with an additive privacy cost while under the local model, a multiplicative privacy cost is shown to be necessary (Garcelon et al., 2021). In addition to valued-based algorithms considered in Vietri et al. (2020); Garcelon et al. (2021), similar performance is established for policy-based algorithms in tabular episodic RL under the central and local model (Chowdhury & Zhou, 2022a). Beyond the tabular setting, differentially private LQR control is studied in Chowdhury et al. (2021). More recently, private episodic RL with linear function approximation has been investigated in Luyo et al. (2021); Zhou (2022); Liao et al. (2021) under both central and local models, where similar regret gap as in contextual bandits exist (i.e., $\widetilde{O}(\sqrt{T})$ vs. $\widetilde{O}(T^{3/4})$).

We have also given more discussions on private (federated) supervised learning under the distributed model in Appendix G. For readers who are interested in the subtlety of DP definitions for bandits and RL, we refer to the blog post (Zhou, 2023).

## B    A POSSIBLE APPROACH TO REDUCE THE COMMUNICATION COST

We first recall that the communication cost for RDP with SecAgg is roughly $O(\log(n+s/\varepsilon))$, where $n$ is the batch size and $s$ is the scaling factor. A large $s$ leads to better privacy and regret as shown in Theorem 2, but incurs a larger communication.

One can observe that the current communication cost is inverse with respect to $\varepsilon$. However, we tend to believe that one can break the privacy-communication trade-off above using a very recent technique proposed in Chen et al. (2022a). In particular, Chen et al. (2022a) shows that the fundamental communication cost for RDP with SecAgg for the mean estimation task scales with $\Omega(\max(\log(n^2\varepsilon^2), 1))$, where $n$ is the batch size. That is, for a stronger privacy guarantee (i.e., a smaller $\varepsilon$), each user should send less number of bits. The intuition is that if a user sends less bits, then she communicates less information about her local data (hence natural protection of privacy).

To achieve this improvement, Chen et al. (2022a) proposes to use a linear compression scheme based on sparse random projections and distributed discrete Gaussian noise. Now, to apply the same technique to the private bandit problem, one needs to handle a different utility metric – that is, instead of the mean-square error in Chen et al. (2022a), one now needs to examine the tail concentration behavior. We leave it as an interesting future work.

## C  A GENERAL REGRET BOUND OF ALGORITHM 1

In this section, we present the proof of our generic regret bound in Lemma 1.

Recall that $n_a(b) := R_a(b) - \sum_{i=1}^{l(b)} r_a^i(b)$ is the total noise injected in the sum of rewards for arm $a$ during batch $b$. We consider the following tail property on $n_a(b)$.

**Assumption 1** (Concentration of Private Noise). *Fix any $p \in (0, 1]$, $a \in [K]$, $b \geq 1$, there exist non-negative constants $\sigma, h$ (possibly depending on $b$) such that, with probability at least $1 - 2p$,*

$$|n_a(b)| \leq \sigma\sqrt{\log(2/p)} + h\log(2/p).$$

We remark that this assumption naturally holds for a single $(\sigma^2, h)$-sub-exponential noise and a single $\sigma^2$-sub-Gaussian noise where $h = 0$ (cf. Lemma 7 and Lemma 6 in Appendix K) with constants adjustment.

**Lemma 2** (Formal statement of Lemma 1). *Let Assumption 1 hold and choose confidence radius*

$$\beta(b) = \sqrt{\frac{\log(4|\Phi(b)|b^2/p)}{2l(b)}} + \frac{\sigma\sqrt{\log(2|\Phi(b)|b^2/p)}}{l(b)} + \frac{h\log(2|\Phi(b)|b^2/p)}{l(b)}, \qquad (2)$$

*where $|\Phi(b)|$ is the number of active arms in batch $b$. Then, for any $p \in (0, 1]$, with probability at least $1 - 3p$, the regret of Algorithm 1 satisfies*

$$Reg(T) = O\left(\sum_{a \in [K]:} \frac{\log(KT/p)}{\Delta_a} + K\sigma\sqrt{\log(KT/p)} + Kh\log(KT/p)\right).$$

*Taking $p = 1/T$ and assuming $T \geq K$, yields the expected regret*

$$\mathbb{E}\left[Reg(T)\right] = O\left(\sum_{a \in [K]:\Delta > 0} \frac{\log T}{\Delta_a} + K\sigma\sqrt{\log T} + Kh\log T\right).$$

*Proof.* Let $\mathcal{E}_b$ be the event that for all active arms $|\widehat{\mu}_a(b) - \mu_a| \leq \beta(b)$ and $\mathcal{E} = \cup_{b \geq 1}\mathcal{E}_b$. Then, we first show that with the choice of $\beta(b)$ given by equation 2, we have $\mathbb{P}[\mathcal{E}] \geq 1 - 3p$ for any $p \in (0, 1]$ under Assumption 1. To see this, we note that

$$\widehat{\mu}_a(b) - \mu_a = \frac{n_a(b) + \sum_{i=1}^{l(b)} r_a^i(b)}{l(b)} - \mu_a.$$

By Hoeffeding's inequality (cf. Lemma 8), we have for any $p \in (0, 1)$, with probability at least $1 - p$,

$$\left|\frac{\sum_{i=1}^{l(b)} r_a^i(b)}{l(b)} - \mu_a\right| = \sqrt{\frac{\log(2/p)}{2l(b)}}.$$

Then, by the concentration of noise $n_a(b)$ in Assumption 1 and triangle inequality, we obtain for a given arm $a$ and batch $b$, with probability at least $1 - 3p$

$$|\widehat{\mu}_a(b) - \mu_a| = \sqrt{\frac{\log(2/p)}{2l(b)}} + \frac{\sigma\sqrt{\log(1/p)}}{l(b)} + \frac{h\log(1/p)}{l(b)}.$$

Thus, by the choice of $\beta(b)$ and a union bound, we have $\mathbb{P}[\mathcal{E}] \geq 1 - 3p$.

In the following, we condition on the good event $\mathcal{E}$. We first show that the optimal arm $a^*$ will always be active. We show this by contradiction. Suppose at the end of some batch $b$, $a^*$ will be

eliminated, i.e., $\text{UCB}_{a^*}(b) < \text{LCB}_{a'}(b)$ for some $a'$. This implies that under good event $\mathcal{E}$

$$\mu_{a^*} \leq \widehat{\mu}_{a^*}(b) + \beta(b) < \widehat{\mu}_{a'}(b) - \beta(b) \leq \mu_{a'},$$

which contradicts the fact that $a^*$ is the optimal arm.

Then, we show that at the end of batch $b$, all arms such that $\Delta_a > 4\beta(b)$ will be eliminated. To show this, we have that under good event $\mathcal{E}$

$$\widehat{\mu}_a(b) + \beta(b) \leq \mu_a(b) + 2\beta(b) < \mu_{a^*}(b) - 4\beta(b) + 2\beta(b) \leq \widehat{\mu}_{a^*}(b) - \beta(b),$$

which implies that arm $a$ will be eliminated by the rule. Thus, for each sub-optimal arm $a$, let $\tilde{b}_a$ be the last batch that arm $a$ is not eliminated. By the above result, we have

$$\Delta_a \leq 4\beta(\tilde{b}_a) = O\left(\sqrt{\frac{\log(KT/p)}{l(\tilde{b}_a)}} + \frac{\sigma\sqrt{\log(KT/p)}}{l(\tilde{b}_a)} + \frac{h\log(KT/p)}{l(\tilde{b}_a)}\right).$$

Hence, we have for some absolute constants $c_1, c_2, c_3$,

$$l(\tilde{b}_a) \leq \max\left\{\frac{c_1\log(KT/p)}{\Delta_a^2}, \frac{c_2\sigma\sqrt{\log(KT/p)}}{\Delta_a}, \frac{c_3 h\log(KT/p)}{\Delta_a}\right\}$$

Since the batch size doubles, we have $N_a(T) \leq 4l(\tilde{b}_a)$ for each sub-optimal arm $a$. Therefore, $\text{Reg}(T) = \sum_{a\in[K]} N_a(T)\Delta_a \leq 4l(\tilde{b}_a)\Delta_a$. Moreover, choose $p = 1/T$ and assume $T \geq K$, we have that the expected regret satisfies

$$\begin{aligned}
\text{Reg}(T) &= \mathbb{E}\left[\sum_{a\in[K]} \Delta_a N_a(T)\right] \\
&\leq \mathbb{P}\left[\bar{\mathcal{E}}\right] \cdot T + O\left(\sum_{a\in[K]:\Delta>0} \frac{\log T}{\Delta_a}\right) + O\left(K\sigma\sqrt{\log T}\right) + O\left(Kh\log T\right) \\
&= O\left(\sum_{a\in[K]:\Delta>0} \frac{\log T}{\Delta_a} + K\sigma\sqrt{\log T} + Kh\log T\right).
\end{aligned}$$

$\square$

**Remark 4.** *In stead of a doubling batch schedule, one can also set $l(b) = \eta^b$ for some absolute constant $\eta > 1$ while attaining the same order of regret bound.*

## D  APPENDIX FOR PURE DP IN SECTION 4

In this section, we provide proofs for Theorem 1 and Theorem 3, which show that pure DP can be achieved under the distributed model via SecAgg and secure shuffling, respectively. Both results build on the generic regret bound in Lemma 2.

### D.1  PROOF OF THEOREM 1

*Proof.* Privacy: We need to show that the server's view at each batch has already satisfies $(\varepsilon, 0)$-DP, which combined with the fact of unique users and parallel composition, yields that Algorithm 1 satisfies $(\varepsilon, 0)$-DP in the distributed model. To this end, in the following, we fix a batch $b$ and arm $a$, and hence $x_i = r_a^i(b)$ and $n = l(b)$. Note that the server's view for each batch is given by

$$\widehat{y} \overset{(a)}{=} \left(\sum_{i\in[n]} y_i\right) \bmod m \overset{(b)}{=} \left(\sum_{i\in[n]} \widehat{x}_i + \eta_i\right) \bmod m, \tag{3}$$

where (a) holds by SecAgg function; (b) holds by the distributive property: $(a + b) \bmod c = (a \bmod c + b \bmod c) \bmod c$ for any $a, b, c \in \mathbb{Z}$. Thus, the view of the server can be simulated as a post-processing of a mechanism $\mathcal{H}$ that accepts an input dataset $\{\widehat{x}_i\}_i$ and outputs $\sum_i \widehat{x}_i + \sum_i \eta_i$. Hence, it suffices to show that $\mathcal{H}$ is $(\varepsilon, 0)$-DP by post-processing of DP. To this end, we note that the sensitivity of $\sum_i \widehat{x}_i$ is $g$, which, by Fact 3, implies that $\sum_i \eta_i$ needs to be distributed as $\mathbf{Lap}_{\mathbb{Z}}(g/\varepsilon)$

in order to guarantee $\varepsilon$-DP. Finally, by Fact 1, it suffices to generate $\eta_i = \gamma_i^+ - \gamma_i^-$, where $\gamma_i^+$ and $\gamma_i^-$ are i.i.d samples from $\textbf{Pólya}(1/n, e^{-\varepsilon/g})$.

Regret: Thanks to the generic regret bound in Lemma 2, we only need to verify Assumption 1. To this end, fix any batch $b$ and arm $a$, we have $\widehat{y} = \widehat{y}_a(b)$, $x_i = r_a^i(b)$ and $n = l(b)$. Then, in the following we will show that with probability at least $1 - 2p$

$$\left| \mathcal{A}(\widehat{y}) - \sum_i x_i \right| \leq O\left( \frac{1}{\varepsilon}\sqrt{\log(1/p)} + \frac{1}{\varepsilon}\log(1/p) \right), \tag{4}$$

which implies that Assumption 1 holds with $\sigma = O(1/\varepsilon)$ and $h = O(1/\varepsilon)$.

Inspired by Cheu & Yan (2021); Balle et al. (2020), we first divide the LHS of equation 4 as follows.

$$\left| \mathcal{A}(\widehat{y}) - \sum_i x_i \right| \leq \underbrace{\left| \mathcal{A}(\widehat{y}) - \frac{1}{g}\sum_i \widehat{x}_i \right|}_{\text{Term (i)}} + \underbrace{\left| \frac{1}{g}\sum_i \widehat{x}_i - \sum_i x_i \right|}_{\text{Term (ii)}},$$

where Term (i) captures the error due to private noise and modular operation, and Term (ii) captures the error due to random rounding.

To start with, we will bound Term (ii). More specifically, we will show that for any $p \in (0, 1]$, with probability at least $1 - p$,

$$\text{Term (ii)} \leq O\left( \frac{1}{\varepsilon}\sqrt{\log(1/p)} \right). \tag{5}$$

Let $\bar{x}_i := \lfloor x_i \cdot g \rfloor$, then $\widehat{x}_i = \bar{x}_i + \textbf{Ber}(x_i g - \bar{x}_i) = x_i g + \bar{x}_i + \textbf{Ber}(x_i g - \bar{x}_i) - x_i g = x_i g + \iota_i$, where $\iota_i := \bar{x}_i + \textbf{Ber}(x_i g - \bar{x}_i) - x_i g$. We have $\mathbb{E}\left[\iota_i\right] = 0$ and $\iota_i \in [-1, 1]$. Hence, $\iota_i$ is 1-sub-Gaussian and as a result, $\frac{1}{g}\sum_i \widehat{x}_i - \sum_i x_i = \frac{1}{g}\sum_i \iota_i$ is $n/g^2$-sub-Gaussian. Therefore, by the concentration of sub-Gaussian (cf. Lemma 6), we have

$$\mathbb{P}\left[ \left| \sum_i \frac{1}{g} \cdot \widehat{x}_i - \sum_i x_i \right| > \sqrt{2\frac{n}{g^2}\log(2/p)} \right] \leq p.$$

Hence, by the choice of $g = \lceil \varepsilon\sqrt{n} \rceil$, we establish equation 5.

Now, we turn to bound Term (i). Recall the choice of parameters: $g = \lceil \varepsilon\sqrt{n} \rceil$, $\tau = \lceil \frac{g}{\varepsilon}\log(2/p) \rceil$, and $m = ng + 2\tau + 1$. We would like to show that

$$\mathbb{P}\left[ \left| \mathcal{A}(\widehat{y}) - \frac{1}{g}\sum_i \widehat{x}_i \right| > \frac{\tau}{g} \right] \leq p, \tag{6}$$

which implies that for any $p \in (0, 1]$, with probability at least $1 - p$

$$\text{Term (i)} \leq O\left( \frac{1}{\varepsilon}\log(2/p) \right). \tag{7}$$

To show equation 6, the key is to bound the error due to private noise and handle the possible underflow carefully. First, we know that the total private noise $\sum_i \eta_i$ is distributed as $\textbf{Lap}_{\mathbb{Z}}(g/\varepsilon)$. Hence, by the concentration of discrete Laplace (cf. Fact 3), we have

$$\mathbb{P}\left[ \left| \sum_i \eta_i \right| > \tau \right] \leq p.$$

Let $\mathcal{E}_{\text{noise}}$ denote the event that $\sum_i \widehat{x}_i + \eta_i \in [\sum_i \widehat{x}_i - \tau, \sum_i \widehat{x}_i + \tau]$, then by the above inequality, we have $\mathbb{P}\left[\mathcal{E}_{\text{noise}}\right] \geq 1 - p$. In the following, we condition on the event of $\mathcal{E}_{\text{noise}}$ to analyze the output $\mathcal{A}(\widehat{y})$. As already shown in equation 3, the input $\widehat{y} = (\sum_i \widehat{x}_i + \eta_i) \mod m$ is already an integer. We let $y = \widehat{y}$. We will consider two cases of $y$ as in the analyzer subroutine $\mathcal{A}$.

**Case 1:** $y > ng + \tau$. We argue that this happens only when $\sum_i \widehat{x}_i + \eta_i \in [-\tau, 0)$, i.e., underflow. This is because for all $i \in [n]$, $\widehat{x}_i \in [0, g]$, $m = ng + 2\tau + 1$ and the total privacy noise is at most $\tau$

under $\mathcal{E}_{\text{noise}}$. Therefore,

$$y - m = \left( \left( \sum_i \widehat{x}_i + \eta_i \right) \bmod m \right) - m$$

$$= \left( m + \sum_i \widehat{x}_i + \eta_i \right) - m$$

$$= \sum_i \widehat{x}_i + \eta_i.$$

That is, $y - m \in [\sum_i \widehat{x}_i - \tau, \sum_i \widehat{x}_i + \tau]$ with high probability. In other words, we have shown that when $y > ng + \tau$, $\mathcal{A}(\widehat{y}) = \frac{y-m}{g}$ satisfies equation 6.

**Case 2:** $y \le ng + \tau$. Here, we have noisy sum $\sum_i \widehat{x}_i + \eta_i \in [0, ng + \tau]$. Hence, $y = \sum_i \widehat{x}_i + \eta_i$ since $m = ng + 2\tau + 1$, which implies that $\mathcal{A}(\widehat{y}) = \frac{y}{g}$ satisfies equation 6.

Hence, we have shown that the output of the analyzer under both cases satisfies equation 6, which implies equation 7. Combined with the bound in equation 5, yields the bound in equation 4. Finally, plugging in $\sigma = O(1/\varepsilon)$, $h = O(1/\varepsilon)$ into the generic regret bound in Lemma 2, yields the required regret bound and completes the proof. □

### D.2 Pure DP via Shuffling

As stated in the main paper (see Remark 1), one can achieve same privacy and regret guarantees (orderwise) using a *relaxed* SecAgg protocol, which relaxes the SecAgg protocol mentioned above in the following sense: (i) **relaxed correctness** – the output of $\mathcal{S}$ can be used to compute the correct modular sum except at most a small probability (denoted by $\widehat{q}$); (ii) **relaxed security** – the output of $\mathcal{S}$ reveals only $\widehat{\varepsilon}$ more information than the modular sum result. Putting the two aspects together, one obtains a relaxed protocol denoted by $(\widehat{\varepsilon}, \widehat{q})$-SecAgg. One important benefit of using this relaxation is that it allows us to achieve the same results of Theorem 1 via secure shuffling. More specifically, as shown in Cheu & Yan (2021), there exists a shuffle protocol that can simulate an $(\widehat{\varepsilon}, \widehat{q})$-SecAgg. Hence, we can directly instantiate $\mathcal{S}$ using this shuffle protocol to achieve pure DP in the distributed model while obtaining the same regret bound as in the central model. We provide more details on this shuffle protocol below.

To facilitate our discussion, we briefly give more formal definitions of a relaxed SecAgg based on Cheu & Yan (2021), which will also be used in our next proof. We denote a perfect SecAgg as $\Sigma$.

To start with, we first need the following two distance metrics between two probability distributions. As in Cheu & Yan (2021), for a given distribution $\mathbf{D}$ and event $E$, we write $\mathbb{P}[\mathbf{D} \in E]$ for $\mathbb{P}_{\eta \sim \mathbf{D}}[\eta \in E]$. We let $\text{supp}(\mathbf{D}, \mathbf{D}')$ be the union of their supports.

**Definition 6** (Statistical Distance). *For any pair of distributions* $\mathbf{D}, \mathbf{D}'$, *the statistical distance is given by*

$$SD(\mathbf{D}, \mathbf{D}') := \max_{E \in supp(\mathbf{D}, \mathbf{D}')} |\mathbb{P}[\mathbf{D} \in E] - \mathbb{P}[\mathbf{D}' \in E]|.$$

**Definition 7** (Log-Likelihood-Ratio (LLR) Distance). *For any pair of distributions* $\mathbf{D}, \mathbf{D}'$, *the LLR distance is given by*

$$LLR(\mathbf{D}, \mathbf{D}') := \max_{E \in supp(\mathbf{D}, \mathbf{D}')} \left| \log \left( \frac{\mathbb{P}[\mathbf{D} \in E]}{\mathbb{P}[\mathbf{D}' \in E]} \right) \right|.$$

Pure DP can be defined using LLR distance.

**Definition 8.** *A randomized mechanism* $\mathcal{M}$ *is* $(\varepsilon, 0)$-*DP if for any two neighboring datasets* $D, D'$, $LLR(\mathcal{M}(D), \mathcal{M}(D')) \le \varepsilon$

**Definition 9** (Relaxed SecAgg). *We say* $\mathcal{S}$ *is an* $(\widehat{\varepsilon}, \widehat{q})$-*relaxation of* $\Sigma$ *(i.e.,* $(\widehat{\varepsilon}, \widehat{q})$-*relaxed SecAgg) if it satisfies the following two conditions for any input* $y$.

1. *($\widehat{q}$-relaxed correctness) There exists some post-processing function* POST *such that* $SD(\text{POST}(\mathcal{S}(y)), \Sigma(y)) \le \widehat{q}$.

> 2. *($\widehat{\varepsilon}$-relaxed security) There exists a simulator $\mathtt{SIM}$ such that $LLR(\mathcal{S}(y), \mathtt{SIM}(\Sigma(y))) \leq \widehat{\varepsilon}$.*

Cheu & Yan (2021) show that one can simulate a relaxed SecAgg via a shuffle protocol. We now briefly talk about the high-level idea behind this idea and refer readers to Cheu & Yan (2021) for details. To simulate an $(\widehat{\varepsilon}, \widehat{q})$-SecAgg via shuffling, the key is to introduce another local randomizer on top of the original $\mathcal{R}$. More specifically, we let $\mathcal{S} := S \circ R^n$ in Algorithm 1 with $n = l(b), b \geq 1$, where $R$ denotes the additional local randomizer at each user $i \in [n]$ that maps the output of $\mathcal{R}$ (i.e., $y_i$) into a random binary vector of a particular length $d$. Then, $S$ denotes a standard shuffler that uniformly at random permutes all the received bits from $n$ users (i.e., a total of $n \cdot d$ bits). The nice construction of $R$ in Cheu & Yan (2021) ensures that $\mathcal{S} = S \circ R^n$ can simulate an $(\widehat{\varepsilon}, \widehat{q})$-SecAgg for any $\widehat{\varepsilon}, \widehat{q} \in (0, 1)$.

The following theorem says that a relaxed SecAgg is sufficient for the same order of expected regret bound while guaranteeing pure DP.

**Theorem 3** (Pure-DP via Shuffling). *Fix $\varepsilon > 0$ and $T \in \mathbb{N}$ and consider an $(\widehat{\varepsilon}, \widehat{q})$-SecAgg (e.g., simulated via shuffling) for Algorithm 1[6]. Let noise for $i$-th user be $\eta_i = \gamma_i^+ - \gamma_i^-$, where $\gamma_i^+, \gamma_i^- \overset{i.i.d.}{\sim}$ Pólya$(1/n, e^{-\varepsilon/g})$. For each batch $b$, choose $n = l(b), g = \lceil \varepsilon' \sqrt{n} \rceil, \tau = \lceil \frac{g}{\varepsilon'} \log(2/p') \rceil$, and $m = ng + 2\tau + 1, \widehat{\varepsilon} = \varepsilon/4, \varepsilon' = \varepsilon/2, \widehat{q} = p' = \frac{1}{2T}$. Then, Algorithm 1 achieves $(\varepsilon, 0)$-DP in the distributed model. Moreover, setting $\beta(b) = O\left( \sqrt{\frac{\log(|\Phi(b)|b^2 T)}{2l(b)}} + \frac{2 \log(|\Phi(b)|b^2 T)}{\varepsilon l(b)} \right)$, it enjoys expected regret*

$$\mathbb{E}\left[Reg(T)\right] = O\left( \sum_{a \in [K]: \Delta_a > 0} \frac{\log T}{\Delta_a} + \frac{K \log T}{\varepsilon} \right).$$

*Moreover, the communication per user before $\mathcal{S}$ is $O(\log m)$ bits.*

*Proof.* This proof shares the same idea as in the proof of Theorem 1. We only need to highlight the difference.

Privacy: As before, we need to ensure that the view of the server already satisfies $\varepsilon$-DP for each batch $b \geq 1$. In the following, we fix any batch $b$ and arm $a$, we have $x_i = r_a^i(b)$ and $n = l(b)$.

Thus, by Definition 8, it suffices to show that for any two neighboring datasets

$$LLR(\mathcal{S}(\mathcal{R}(x_1), \ldots, \mathcal{R}(x_n)), \mathcal{S}(\mathcal{R}(x_1'), \ldots, \mathcal{R}(x_n'))) \leq \varepsilon.$$

To this end, by the $\widehat{\varepsilon}$-relaxed security of $\mathcal{S}$ used in Algorithm 1 and triangle inequality, we have

$$LLR(\mathcal{S}(\mathcal{R}(x_1), \ldots, \mathcal{R}(x_n)), \mathcal{S}(\mathcal{R}(x_1'), \ldots, \mathcal{R}(x_n')))$$

$$\leq 2\widehat{\varepsilon} + LLR(\mathtt{SIM}(\sum_i \mathcal{R}(x_i) \bmod m), \mathtt{SIM}(\sum_i \mathcal{R}(x_i') \bmod m))$$

$$\leq 2\widehat{\varepsilon} + LLR(\sum_i \mathcal{R}(x_i) \bmod m, \sum_i \mathcal{R}(x_i') \bmod m),$$

where the last inequality follows from data processing inequality. The remaining step to bound the second term above is the same as in the proof of Theorem 1. With $\widehat{\varepsilon} = \varepsilon/4$ and $\varepsilon' = \varepsilon/2$, we have the total privacy loss is $\varepsilon$.

Regret: As before, the key is to establish that with high probability

$$\left| \mathcal{A}(\widehat{y}) - \sum_i x_i \right| \leq O\left( \frac{1}{\varepsilon} \sqrt{\log(2/p')} + \frac{1}{\varepsilon} \log(2/p') \right), \tag{8}$$

where $\widehat{y} := \widehat{y}_a(b)$.

---

[6]Since the output of a shuffling protocols is a multiset, we need to first compute $(\sum \widehat{y}) \bmod m$ as the $\widehat{y}$ for the subroutine $\mathcal{A}$ in Algorithm 1.

We again can divide the LHS of equation 8 into

$$\left| \mathcal{A}(\widehat{y}) - \sum_i x_i \right| \leq \underbrace{\left| \mathcal{A}(\widehat{y}) - \frac{1}{g}\sum_i \widehat{x}_i \right|}_{\text{Term (i)}} + \underbrace{\left| \frac{1}{g}\sum_i \widehat{x}_i - \sum_i x_i \right|}_{\text{Term (ii)}}.$$

In particular, Term (ii) can be bounded by using the same method, i.e., for any $p' \in (0,1]$, with probability at least $1 - p'$,

$$\text{Term (ii)} \leq O\left( \frac{1}{\varepsilon'}\sqrt{\log(2/p')} \right).$$

For Term (i), we would like to show that

$$\mathbb{P}\left[ \left| \mathcal{A}(\widehat{y}) - \frac{1}{g}\sum_i \widehat{x}_i \right| > \frac{\tau}{g} \right] \leq \widehat{q} + p'.$$

This can be established by the same steps in the proof of Theorem 1 while conditioning on the high probability event that $\mathcal{S}$ is $\widehat{q}$-relaxed SecAgg. More specifically, compared to the proof of Theorem 1, the key difference here is that $\widehat{y} = (\sum_i y_i) \mod m$ only holds with high probability $1 - \widehat{q}$ by the definition of $\widehat{q}$-relaxed SecAgg.

Thus, for any $p' \in (0,1)$, let $\widehat{q} = p'$, we have with probability at least $1 - 3p'$, $|\mathcal{A}(\widehat{y}) - \sum_i x_i| \leq O\left( \frac{1}{\varepsilon}\sqrt{\log(2/p')} + \frac{1}{\varepsilon}\log(2/p') \right)$, which implies that Assumption 1 holds with $\sigma = O(1/\varepsilon)$ and $h = O(1/\varepsilon)$. $\qquad\square$

## E  APPENDIX FOR RDP IN SECTION 5

### E.1  PROOF OF PROPOSITION 1

*Proof.* We first establish the following result.

**Claim 1.** *For all $\lambda \in \mathbb{R}$, we have*

$$\cosh(\lambda) \leq e^{\lambda^2/2}.$$

To show this, by the infinite product representation of the hyperbolic cosine function, we have

$$\cosh(\lambda) = \prod_{k=1}^{\infty}\left( 1 + \frac{4\lambda^2}{\pi^2(2k-1)^2} \right) \overset{(a)}{\leq} \exp\left( \sum_{k=1}^{\infty} \frac{4\lambda^2}{\pi^2(2k-1)^2} \right) \overset{(b)}{=} \exp(\lambda^2/2),$$

where (a) holds by the fact that $1 + x \leq e^x$ for all $x \in \mathbb{R}$ and (b) follows from $\sum_{k=1}^{\infty} \frac{1}{(2k-1)^2} = \frac{\pi^2}{8}$.

Then, note that $X \sim \mathbf{Sk}(0, \sigma^2)$, then its moment generating function (MGF) is given by $\mathbb{E}\left[ e^{\lambda X} \right] = \exp(\sigma^2(\cosh(\lambda) - 1))$. Hence, by the above claim, we have $\mathbb{E}\left[ e^{\lambda X} \right] \leq \exp(\sigma^2(e^{\lambda^2/2} - 1))$. Further, note that $e^x - 1 \leq 2x$ for $x \in [0, 1]$. Thus, for $|\lambda| \leq \sqrt{2}$, we have

$$\mathbb{E}\left[ e^{\lambda X} \right] \leq e^{\lambda^2 \sigma^2} = e^{\frac{\lambda^2 2\sigma^2}{2}}.$$

Hence, by the definition of sub-exponential random variable (cf. Lemma 7), $X$ is $(2\sigma^2, \frac{\sqrt{2}}{2})$-sub-exponential, which again by Lemma 7 implies the required concentration result, i.e., for any $p \in (0, 1]$, with probability at least $1 - p$,

$$|X| \leq 2\sigma\sqrt{\log(2/p)} + \sqrt{2}\log(2/p).$$

$\qquad\square$

### E.2  PROOF OF THEOREM 2

We will leverage the following result in (Agarwal et al., 2021, Theorem 3.5 ) to prove privacy guarantee.

**Lemma 3.** *For $\alpha \in \mathbb{Z}$, $\alpha > 1$, let $X \sim \mathbf{Sk}(0, \sigma^2)$. Then, an algorithm $M$ that adds $X$ to a sensitivity-$\Delta$ query satisfies $(\alpha, \varepsilon(\alpha))$-RDP with $\varepsilon(\alpha)$ given by*

$$\varepsilon(\alpha) \leq \frac{\alpha \Delta^2}{2\sigma^2} + \min\left\{\frac{(2\alpha - 1)\Delta^2 + 6\Delta}{4\sigma^4}, \frac{3\Delta}{2\sigma^2}\right\}.$$

*Proof of Theorem 2.* Privacy: As before, we fix any batch $b$ and arm $a$, for simplicity, we let $x_i = r_a^i(b)$ and $n = l(b)$. Then, it suffices to show that the mechanism $\mathcal{H}$ that accepts an input dataset $\{\widehat{x}_i\}_i$ and outputs $\sum_i \widehat{x}_i + \sum_i \eta_i$ is private. To this end, since each local randomizer $\mathcal{R}$ generates noise $\eta_i \sim \mathbf{Sk}(0, \frac{g^2}{n\varepsilon^2})$ and Skellam is closed under summation, we have that the total noise $\sum_i \eta_i \sim \mathbf{Sk}(0, \frac{g^2}{\varepsilon^2})$. Thus, by Lemma 3 with $\Delta = g$, we have that for each batch $b$ with $n = l(b)$ and $g = \lceil s\varepsilon\sqrt{n}\rceil$, Algorithm 1 is $(\alpha, \widehat{\varepsilon}_n(\alpha))$-RDP with $\widehat{\varepsilon}_n(\alpha)$ given by

$$\widehat{\varepsilon}_n(\alpha) = \frac{\alpha \varepsilon^2}{2} + \min\left\{\frac{(2\alpha - 1)\varepsilon^2}{4s^2n} + \frac{3\varepsilon}{2s^3n^{3/2}}, \frac{3\varepsilon^2}{2s\sqrt{n}}\right\}.$$

Since $n = l(b) > 1$, we have that for all batches $b \geq 1$,

$$\widehat{\varepsilon}_n(\alpha) \leq \widehat{\varepsilon}(\alpha) := \frac{\alpha \varepsilon^2}{2} + \min\left\{\frac{(2\alpha - 1)\varepsilon^2}{4s^2} + \frac{3\varepsilon}{2s^3}, \frac{3\varepsilon^2}{2s}\right\}.$$

Regret: We will establish the following high probability bound so that we can apply our generic regret bound in Lemma 2

$$\left|\mathcal{A}(\widehat{y}) - \sum_i x_i\right| \leq O\left(\frac{1}{\varepsilon}\sqrt{\log(1/p)} + \frac{1}{s\varepsilon}\log(2/p)\right), \tag{9}$$

where $\widehat{y} := \widehat{y}_a(b)$ for each batch $b$ and arm $a$.

We again divide the LHS of equation 9 into

$$\left|\mathcal{A}(\widehat{y}) - \sum_i x_i\right| \leq \underbrace{\left|\mathcal{A}(\widehat{y}) - \frac{1}{g}\sum_i \widehat{x}_i\right|}_{\text{Term (i)}} + \underbrace{\left|\frac{1}{g}\sum_i \widehat{x}_i - \sum_i x_i\right|}_{\text{Term (ii)}}.$$

In particular, Term (ii) can be bounded by using the same method before, i.e.,

$$\mathbb{P}\left[\left|\sum_i \frac{1}{g}\cdot\widehat{x}_i - \sum_i x_i\right| > \sqrt{2\frac{n}{g^2}\log(2/p)}\right] \leq p.$$

Hence, by the choice of $g = \lceil s\varepsilon\sqrt{n}\rceil$, we establish that with high probability

$$\text{Term (ii)} \leq O\left(\frac{1}{\varepsilon \cdot s}\sqrt{\log(1/p)}\right).$$

For Term (i), as in the previous proof, the key is to show that

$$\mathbb{P}\left[\left|\sum_{i\in[n]} \eta_i\right| > \tau\right] \leq p.$$

To this end, we will utilize our established result in Proposition 1. Note that the total noise $\sum_i \eta_i \sim \mathbf{Sk}(0, \frac{g^2}{\varepsilon^2})$, and hence by Proposition 1 and the choice of $\tau = \lceil\frac{2g}{\varepsilon}\sqrt{\log(2/p)} + \sqrt{2}\log(2/p)\rceil$, we have the above result. Following previous proof, this result implies that with high probability

$$\text{Term (i)} \leq \frac{\tau}{g} \leq O\left(\frac{1}{\varepsilon}\sqrt{\log(1/p)} + \frac{1}{s\varepsilon}\log(1/p)\right).$$

Combining the bounds on Term (i) and Term (ii), we have that the private noise satisfies Assumption 1 with constants $\sigma = O(1/\varepsilon)$ and $h = O(\frac{1}{s\varepsilon})$. Hence, by the generic regret bound in Lemma 2, we have established the required result. $\qquad\square$

# F ACHIEVING CDP IN THE DISTRIBUTED MODEL VIA DISCRETE GAUSSIAN

We first introduce the following definition of concentrated differential privacy Bun & Steinke (2016).

**Definition 10** (Concentrated Differential Privacy). *A randomized mechanism $\mathcal{M}$ satisfies $\frac{1}{2}\varepsilon^2$-CDP if for all neighboring datasets $D, D'$ and all $\alpha \in (1, \infty)$, we have $D_\alpha(\mathcal{M}(D), \mathcal{M}(D')) \leq \frac{1}{2}\varepsilon^2\alpha$.*

**Remark 5** (CDP vs. RDP). *From the definition, we can see that $\frac{1}{2}\varepsilon^2$-CDP is equivalent to satisfying $(\alpha, \frac{1}{2}\varepsilon^2\alpha)$-RDP simultaneously for all $\alpha > 1$.*

The discrete Gaussian mechanism is first proposed and investigated by Canonne et al. (2020) and has been recently applied to federated learning Kairouz et al. (2021a). We apply it to bandit learning to demonstrate the flexibility of our proposed algorithm and analysis.

**Definition 11** (Discrete Gaussian Distribution). *Let $\mu, \sigma \in \mathbb{R}$ with $\sigma > 0$. A random variable $X$ has a discrete Gaussian distribution with location $\mu$ and scale $\sigma$, denoted by $\mathcal{N}_{\mathbb{Z}}(\mu, \sigma^2)$, if it has a probability mass function given by*

$$\forall x \in \mathbb{Z}, \quad \mathbb{P}[X = x] = \frac{e^{-(x-\mu)^2/2\sigma^2}}{\sum_{y \in \mathbb{Z}} e^{-(y-\mu)^2/2\sigma^2}}.$$

The following result from Canonne et al. (2020) will be useful in our privacy and regret analysis.

**Fact 2** (Discrete Gaussian Privacy and Utility). *Let $\Delta, \varepsilon > 0$. Let $q : \mathcal{X}^n \to \mathbb{Z}$ satisfy $|q(x) - q(x')| \leq \Delta$ for all $x, x'$ differing on a single entry. Define a randomized algorithm $M : \mathcal{X}^n \to \mathbb{Z}$ by $M(x) = q(x) + Y$, where $Y \sim \mathcal{N}_{\mathbb{Z}}(0, \Delta^2/\varepsilon^2)$. Then, $M$ satisfies $\frac{1}{2}\varepsilon^2$-concentrated differential privacy (CDP). Moreover, $Y$ is $\Delta^2/\varepsilon^2$-sub-Gaussian and hence for all $t \geq 0$,*

$$\mathbb{P}[|Y| \geq t] \leq 2e^{-t^2\varepsilon^2/(2\Delta^2)}.$$

However, a direct application of above results does not work. This is because the sum of discrete Gaussian is *not* a discrete Gaussian, and hence one cannot directly apply the privacy guarantee of discrete Gaussian when analyzing the view of the analyzer via summing all the noise from users. To overcome this, we will rely on a recent result in Kairouz et al. (2021a), which shows that under reasonable parameter regimes, the sum of discrete Gaussian is close to a discrete Gaussian. The regret analysis will again build on the generic regret bound for Algorithm 1.

**Theorem 4** (CDP via SecAgg). *Fix $\varepsilon \in (0, 1)$ and a scaling factor $s \geq 1$. Let noise for $i$-th user be $\eta_i \sim \mathcal{N}_{\mathbb{Z}}(0, \frac{g^2}{n\varepsilon^2})$. For each $b \geq 1$, choose $n = l(b), g = \lceil s\varepsilon\sqrt{n} \rceil, \tau = \lceil \frac{g}{\varepsilon}\sqrt{2\log(2/p)} \rceil, p = 1/T$ and $m = ng + 2\tau + 1$. Then, Algorithm 1 achieves $\frac{1}{2}\widehat{\varepsilon}^2$-CDP with $\widehat{\varepsilon}$ given by*

$$\widehat{\varepsilon} \leq \min\left\{\sqrt{\varepsilon^2 + \frac{1}{2}\xi}, \varepsilon + \xi\right\},$$

*where $\xi := 10 \cdot \sum_{k=1}^{\frac{T}{2}-1} e^{-2\pi^2 s^2 \cdot \frac{k}{k+1}}$. Meanwhile, the regret is given by*

$$Reg(T) = O\left(\sum_{a \in [K]:\Delta > 0} \frac{\log T}{\Delta_a} + \frac{K\sqrt{\log T}}{\varepsilon}\right),$$

*and the communication messages per user before $\mathcal{S}$ are $O(\log m)$ bits.*

**Remark 6** (Privacy-Communication Trade-off). *We can observe an interesting trade-off between privacy and communication cost. In particular, as $s$ increases, $\xi$ approaches zero and hence the privacy loss approaches the one under continuous Gaussian. However, a larger $s$ leads to a larger $m$ and hence a larger communication overhead.*

We will leverage the following result in (Kairouz et al., 2021a, Proposition 13) to prove privacy.

**Lemma 4** (Privacy for the sum of discrete Gaussian). *Let $\sigma \geq 1/2$. Let $X_i \sim \mathcal{N}_{\mathbb{Z}}(0, \sigma^2)$ independently for each $i$. Let $Z_n = \sum_{i=1}^n X_i$. Then, an algorithm $M$ that adds $Z_n$ to a sensitivity-$\Delta$ query satisfies $\frac{1}{2}\varepsilon^2$-CDP with $\varepsilon$ given by*

$$\varepsilon = \min\left\{\sqrt{\frac{\Delta^2}{n\sigma^2} + \frac{1}{2}\xi}, \frac{\Delta}{\sqrt{n}\sigma} + \xi\right\},$$

*where* $\xi := 10 \cdot \sum_{k=1}^{n-1} e^{-2\pi^2 \sigma^2 \frac{k}{k+1}}$.

*Proof of Theorem 4.* Privacy: As before, we fix any batch $b$ and arm $a$, for simplicity, we let $x_i = r_a^i(b)$ and $n = l(b)$. Then, it suffices to show that the mechanism $\mathcal{H}$ that accepts an input dataset $\{\widehat{x}_i\}_i$ and outputs $\sum_i \widehat{x}_i + \sum_i \eta_i$ is private. To this end, each local randomizer $\mathcal{R}$ generates noise $\eta_i \sim \mathcal{N}_{\mathbb{Z}}(0, \frac{g^2}{n\varepsilon^2})$, and hence by Lemma 4 with $\Delta = g$, we have that $\mathcal{H}$ is $\frac{1}{2}\widehat{\varepsilon}_n^2$-concentrated differential privacy with $\widehat{\varepsilon}_n$ given by

$$\widehat{\varepsilon}_n = \min \left\{ \sqrt{\varepsilon^2 + \frac{1}{2}\xi_n}, \varepsilon + \xi_n \right\},$$

where $\xi_n := 10 \cdot \sum_{k=1}^{n-1} e^{-2\pi^2 \frac{g^2}{n\varepsilon^2} \cdot \frac{k}{k+1}}$. Note that $g = \lceil s\varepsilon\sqrt{n} \rceil$, we have $\tau_n \leq \tau := 10 \cdot \sum_{k=1}^{\frac{T}{2}-1} e^{-2\pi^2 s^2 \cdot \frac{k}{k+1}}$ since $n = l(b) \leq T/2$. Thus, we have that Algorithm 1 is is $\frac{1}{2}\widehat{\varepsilon}^2$-concentrated differential privacy with $\widehat{\varepsilon}$ given by

$$\widehat{\varepsilon} = \min \left\{ \sqrt{\varepsilon^2 + \frac{1}{2}\xi}, \varepsilon + \xi \right\},$$

where $\xi := 10 \cdot \sum_{k=1}^{\frac{T}{2}-1} e^{-2\pi^2 s^2 \cdot \frac{k}{k+1}}$.

Regret: We will establish the following high probability bound so that we can apply our generic regret bound in Lemma 2

$$\left| \mathcal{A}(\widehat{y}) - \sum_i x_i \right| \leq O\left( \frac{1}{\varepsilon} \sqrt{\log(1/p)} \right), \tag{10}$$

where $\widehat{y} := \widehat{y}_a(b)$.

We again divide the LHS of equation 10 into

$$\left| \mathcal{A}(\widehat{y}) - \sum_i x_i \right| \leq \underbrace{\left| \mathcal{A}(\widehat{y}) - \frac{1}{g} \sum_i \widehat{x}_i \right|}_{\text{Term (i)}} + \underbrace{\left| \frac{1}{g} \sum_i \widehat{x}_i - \sum_i x_i \right|}_{\text{Term (ii)}}.$$

In particular, Term (ii) can be bounded by using the same method before, i.e.,

$$\mathbb{P}\left[ \left| \sum_i \frac{1}{g} \cdot \widehat{x}_i - \sum_i x_i \right| > \sqrt{2\frac{n}{g^2} \log(2/p)} \right] \leq p.$$

Hence, by the choice of $g = \lceil s\varepsilon\sqrt{n} \rceil$, we establish that with high probability

$$\text{Term (ii)} \leq O\left( \frac{1}{\varepsilon \cdot s} \sqrt{\log(1/p)} \right).$$

For Term (i), as in the previous proof, the key is to show that

$$\mathbb{P}\left[ \left| \sum_{i \in [n]} \eta_i \right| > \tau \right] \leq p.$$

This follows from the fact that discrete Gaussian is sub-Gaussian and sum of sub-Gaussian is still sub-Gaussian. Thus, by Lemma 6 and the choice of $\tau = \frac{g}{\varepsilon}\sqrt{2\log(2/p)}$, we have the above result, which implies that with high probability

$$\text{Term (i)} \leq \frac{\tau}{g} = O\left( \frac{1}{\varepsilon} \sqrt{\log(1/p)} \right).$$

Since $s \geq 1$, combining the bounds on Term (i) and Term (ii), we have that the private noise satisfies Assumption 1 with constants $\sigma = O(1/\varepsilon)$ and $h = 0$. Hence, by the generic regret bound in Lemma 2, we have established the required result. $\square$

## G   MORE DETAILS ON SECURE AGGREGATION AND SECURE SHUFFLING

In this section, we provide more details about secure aggregation (SecAgg) and secure shuffling including practical implementations and recent theoretical results for DP guarantees. We will also discuss some limitations of both protocols, which in turn highlights that both have some advantages over the other and thus how to select them really depends on particular applications.

### G.1   SECURE AGGREGATION

Practical implementations: SecAgg is a lightweight instance of secure multi-party computation (MPC) based on cryptographic primitives such that the server only learns the aggregated result (e.g., sum) of all participating users' values. This is often achieved via additive masking over a finite group (Bonawitz et al., 2017; Bell et al., 2020). The high-level idea is that participating users add randomly sampled zero-sum mask values by working in the space of integers modulo $m$, which guarantees that each user's masked value is indistinguishable from a random value. However, when all masked values are summed modulo $m$ by the server, all the masks will be cancelled out and the server observes the true modulo sum. Bonawitz et al. (2017) proposed the first scalable SecAgg protocol where both communication and computation costs of each user scale linearly with the number of all participating users. Recently, Bell et al. (2020) presents a further improvement where both client computation and communication depend *logarithmically* on the number of participating clients.

SecAgg for distributed DP: First note that the fact that the server only learns the sum of values under SecAgg does not necessarily imply differential privacy since this aggregated result still has the risk of leaking each user's sensitive information. To provide a formal DP guarantee under SecAgg, each participating user can first perturb her own data with a moderate random noise such that in the aggregated sum, the total noise is large enough to provide a high privacy guarantee. Only until recently, SecAgg with DP has been systematically studied, mainly in the context of private (federated) supervised learning (Kairouz et al., 2021a; Agarwal et al., 2021) while SecAgg in the context of private online learning remains open until our work. More importantly, there exist no formal convergence guarantees of SGD in Kairouz et al. (2021a); Agarwal et al. (2021) due to the biased gradient estimates. To the best of our knowledge, the very recent work (Chen et al., 2022b) is the only one that derives upper bound on the convergence rate when working with SecAgg in private supervised learning. However, the privacy guarantee in it is only approximated DP rather than pure DP considered in our paper.

Limitations of SecAgg: As pointed out in Kairouz et al. (2021b), several limitations of SecAgg still exist despite of recent advances. For example, it assumes a semi-honest server and allows it to see the per-round aggregates. Moreover, it is not efficient for sparse vector aggregation.

### G.2   SECURE SHUFFLING

Practical implementations: Secure shuffling ensures that the server only learns an unordered collection (i.e., multiset) of the messages sent by all the participating users. This is often achieved by a third party shuffling function via cryptographic onion routing or mixnets (Dingledine et al., 2004) or oblivious shuffling with a trusted hardware (Bittau et al., 2017).

Secure shuffling for distributed DP: The additional randomness introduced by the shuffling function can be utilized to achieve a similar utility as the central model while without a trustworthy server. This particular distributed model via secure shuffling is also often called *shuffle model*. In particular, Cheu et al. (2019) first show that for the problem of summing $n$ real-valued numbers within $[0, 1]$, the expected error under shuffle model with $(\varepsilon, \delta)$-DP guarantee is $O(\frac{1}{\varepsilon} \log \frac{n}{\delta})$. For comparison, under the central model, the standard Laplace mechanism achieves $(\varepsilon, 0)$-DP (i.e., pure DP) with an error $O(1/\varepsilon)$ (Dwork et al., 2006) while an error $\Omega(\sqrt{n}/\varepsilon)$ is necessary under the local model (Chan et al., 2012; Beimel et al., 2008). Subsequent works on private scalar sum under the shuffle model have improved both the communication cost and accuracy in Cheu et al. (2019). More specifically, instead of sending $O(\varepsilon\sqrt{n})$ bits per user as in Cheu et al. (2019), the protocols proposed in Balle et al. (2020); Ghazi et al. (2020b) achieve $(\varepsilon, \delta)$-DP with an error $O(1/\varepsilon)$ while each user only sends $O(\log n + \log(1/\delta))$ bits. A closely related direction in the shuffle model is privacy amplification bounds (Erlingsson et al., 2019; Balle et al., 2019; Feldman et al., 2022). That is,

---

**Algorithm 2** Local Randomizer $\mathcal{R}$ (Central Model)

---

1: **Input:** Each user data $x_i \in [0, 1]$
2: **Parameters:** precision $g \in \mathbb{N}$, modulo $m \in \mathbb{N}$, batch size $n \in \mathbb{N}$, privacy parameter $\phi$
3: Encode $x_i$ as $\widehat{x}_i = \lfloor x_i g \rfloor + \mathbf{Ber}(x_i g - \lfloor x_i g \rfloor)$
4: Modulo clip $y_i = \widehat{x}_i \bmod m$ // no local noise
5: **Output:** $y_i$

---

the shuffling of $n$ locally private data yields a gain in privacy guarantees. In particular, Feldman et al. (2022)[7] show that randomly shuffling of $n$ $\varepsilon_0$-DP locally randomized data yields an $(\varepsilon, \delta)$-DP guarantee with $\varepsilon = O\left(\varepsilon_0 \frac{\sqrt{\log(1/\delta)}}{\sqrt{n}}\right)$ when $\varepsilon_0 \leq 1$ and $\varepsilon = O\left(\frac{\sqrt{e^{\varepsilon_0} \log(1/\delta)}}{\sqrt{n}}\right)$ when $\varepsilon_0 > 1$.

Shuffle model has also been studied in the context of empirical risk minimization (ERM) (Girgis et al., 2021) and stochastic convex optimization (Cheu et al., 2021; Lowy & Razaviyayn, 2021), in an attempt to recover some of the utility under the central model while without a trusted server. To the best of our knowledge, Tenenbaum et al. (2021); Chowdhury & Zhou (2022b); Garcelon et al. (2022) are the only works that study shuffle model in the context of bandit learning.

In all the above mentioned works on shuffle model, only approximate DP is achieved. To the best of our knowledge, there are only two existing shuffling protocols that can attain pure DP (Ghazi et al., 2020a; Cheu & Yan, 2021). In particular, by simulating a relaxed SecAgg via a shuffle protocol, Cheu & Yan (2021) are the first to show that there exists a shuffle protocol for bounded sums that satisfies $(\varepsilon, 0)$-DP with an expected error of $O(1/\varepsilon)$. This is main inspiration for us to achieve pure DP in MAB with secure shuffling.

Limitations of secure shuffling: As pointed out in Kairouz et al. (2021b), one of the limitations is the requirement of a trusted intermediary for the shuffling function. Another is that the privacy guarantee under the shuffle model degrades in proportion to the number of adversarial users.

## H    ALGORITHM 1 UNDER CENTRAL AND LOCAL MODELS

In this section, we will show that Algorithm 1 and variants of our template protocol $\mathcal{P}$ allow us to achieve central and local DP using only *discrete noise* while attaining the same optimal regret bound as in previous works using continuous noise (Sajed & Sheffet, 2019; Ren et al., 2020).

### H.1    CENTRAL MODEL

For illustration, we will mainly consider $\mathcal{S}$ as a SecAgg, while secure shuffling can be applied using the same way as in the main paper. In the central model, we just need to set $\eta_i = 0$ in the local randomizer $\mathcal{R}$ and add central noise in the analyzer $\mathcal{A}$ (see Algorithm 2 and Algorithm 3). Then, we have the following privacy and regret guarantees.

**Theorem 5** (Central Pure-DP via SecAgg). *Let $\mathcal{P} = (\mathcal{R}, \mathcal{S}, \mathcal{A})$ be a protocol such that $\mathcal{R}$ given by Algorithm 2, $\mathcal{S}$ is any SecAgg protocol and $\mathcal{A}$ is given by Algorithm 3 with $\eta \sim \mathbf{Lap}_{\mathbb{Z}}(g/\varepsilon)$. Fix any $\varepsilon \in (0, 1)$, for each $b \geq 1$, choose $n = l(b), g = \lceil \varepsilon\sqrt{n} \rceil, \tau = \lceil \frac{g}{\varepsilon} \log(2/p) \rceil, p = 1/T$ and $m = ng + 2\tau + 1$. Then, Algorithm 1 instantiated with protocol $\mathcal{P}$ is able to achieve $(\varepsilon, 0)$-DP in the central model with expected regret given by*

$$\mathbb{E}\left[Reg(T)\right] = O\left(\sum_{a \in [K]:\Delta_a > 0} \frac{\log T}{\Delta_a} + \frac{K \log T}{\varepsilon}\right).$$

*Moreover, the communication messages per user before $\mathcal{S}$ are $O(\log m)$ bits.*

**Remark 7.** *We can generate $\eta \sim \mathbf{Lap}_{\mathbb{Z}}(g/\varepsilon)$ as $\eta = \eta_1 - \eta_2$, where $\eta_1, \eta_2$ is independent and $\mathbf{polya}(1, e^{-\varepsilon/g})$ distributed.*

---

[7]The results in Feldman et al. (2022) hold for adaptive randomizers, but then it needs first to shuffle and then apply the local randomizer. For a fixed randomizer, shuffle-then-randomize is equivalent to randomize-then shuffle.

---

**Algorithm 3** Analyzer $\mathcal{A}$ (Central Model)

1: **Input:** $\widehat{y}$ (output of SecAgg)
2: **Parameters:** precision $g \in \mathbb{N}$, modulo $m \in \mathbb{N}$, batch size $n \in \mathbb{N}$, accuracy parameter $\tau$
3: Generate discrete Laplace noise $\eta$ and set $\widehat{\eta} = \eta \bmod m$
4: Set $y = (\widehat{y} + \widehat{\eta}) \bmod m$
5: **if** $y > ng + \tau$ **then**
6:     set $z = (y - m)/g$ // correction for underflow
7: **else** set $z = y/g$
8: **Output:** $z$

---

*Proof of Theorem 5.* Privacy: By the definition of central DP and post-processing, it suffices to show that $y$ in Algorithm 3 is private. To this end, we again apply the distributive property of modular sum to obtain that

$$y = (\widehat{y} + (\eta \bmod m)) \bmod m = \left( \left( \sum_i \widehat{x}_i \right) + \eta \right) \bmod m.$$

Since the sensitivity of $\sum_i \widehat{x}_i$ is $g$ and $\eta \sim \mathbf{Lap}_{\mathbb{Z}}(g/\varepsilon)$, by Fact 3, we have obtained $(\varepsilon, 0)$-DP in the central model.

Regret: As before, the key is to establish that with high probability

$$\left| \mathcal{A}(\widehat{y}) - \sum_i x_i \right| \leq O\left( \frac{1}{\varepsilon} \sqrt{\log(1/p)} + \frac{1}{\varepsilon} \log(1/p) \right), \tag{11}$$

where $\widehat{y} := \widehat{y}_a(b)$. Then, we can apply our generic regret bound in Lemma 2.

We again can divide the LHS of equation 11 into

$$\left| \mathcal{A}(\widehat{y}) - \sum_i x_i \right| \leq \underbrace{\left| \mathcal{A}(\widehat{y}) - \frac{1}{g} \sum_i \widehat{x}_i \right|}_{\text{Term (i)}} + \underbrace{\left| \frac{1}{g} \sum_i \widehat{x}_i - \sum_i x_i \right|}_{\text{Term (ii)}}.$$

In particular, Term (ii) can be bounded by using the same method, i.e., for any $p \in (0, 1]$, with probability at least $1 - p$,

$$\text{Term (ii)} \leq O\left( \frac{1}{\varepsilon} \sqrt{\log(1/p)} \right).$$

For Term (i), we would like to show that

$$\mathbb{P}\left[ \left| \mathcal{A}(\widehat{y}) - \frac{1}{g} \sum_i \widehat{x}_i \right| > \frac{\tau}{g} \right] \leq p.$$

As before, let $\mathcal{E}_{\text{noise}}$ denote the event that $(\sum_i \widehat{x}_i) + \eta \in [\sum_i \widehat{x}_i - \tau, \sum_i \widehat{x}_i + \tau]$, then by the concentration of discrete Laplace, we have $\mathbb{P}[\mathcal{E}_{\text{noise}}] \geq 1 - p$. In the following, we condition on the event of $\mathcal{E}_{\text{noise}}$ to analyze the output $\mathcal{A}(\widehat{y})$. Note that $y = ((\sum_i \widehat{x}_i) + \eta) \bmod m$ and hence we can use the same steps as in the proof of Theorem 1 to conclude that

$$\text{Term (i)} \leq O\left( \frac{1}{\varepsilon} \log(1/p) \right). \tag{12}$$

Finally, by our generic regret bound in Lemma 2, we have the regret bound. □

## H.2   Local Model

In the local model, each local randomizer $\mathcal{R}$ needs to inject discrete Laplace noise to guarantee pure DP while the analyzer $\mathcal{A}$ is simply the same as in the main paper (see Algorithm 4 and Algorithm 5).

To analyze the regret, we need the concentration of the sum of discrete Laplace. To this end, we have the following result.

**Lemma 5** (Concentration of discrete Laplace). *Let $\{X_i\}_{i=1}^n$ be i.i.d random variable distributed according to $\mathbf{Lap}_{\mathbb{Z}}(1/\varepsilon)$. Then, we have $X_i$ is $(c^2 \frac{1}{\varepsilon^2}, c\frac{1}{\varepsilon})$-sub-exponential for some absolute con-*

---

**Algorithm 4** Local Randomizer $\mathcal{R}$ (Local Model)

---

1: **Input:** Each user data $x_i \in [0, 1]$
2: **Parameters:** precision $g \in \mathbb{N}$, modulo $m \in \mathbb{N}$, batch size $n \in \mathbb{N}$, privacy parameter $\phi$
3: Encode $x_i$ as $\widehat{x}_i = \lfloor x_i g \rfloor + \textbf{Ber}(x_i g - \lfloor x_i g \rfloor)$
4: Generate discrete Laplace noise $\eta_i$
5: Add noise and modulo clip $y_i = (\widehat{x}_i + \eta_i) \bmod m$
6: **Output:** $y_i$ // for SecAgg $\mathcal{S}$

---

**Algorithm 5** Analyzer $\mathcal{A}$ (Local Model)

---

1: **Input:** $\widehat{y}$ (output of SecAgg)
2: **Parameters:** precision $g \in \mathbb{N}$, modulo $m \in \mathbb{N}$, batch size $n \in \mathbb{N}$, accuracy parameter $\tau$
3: Set $y = \widehat{y}$
4: **if** $y > ng + \tau$ **then**
5:     set $z = (y - m)/g$ // correction for underflow
6: **else** set $z = y/g$
7: **Output:** $z$

---

*stant $c > 0$. As a result, we have for any $p \in (0, 1]$, with probability at least $1 - p$, $|\sum_{i=1}^{n} X_i| \le v$, where $v \ge \max\{\frac{c}{\varepsilon}\sqrt{2n \log(2/p)}, 2\frac{c}{\varepsilon}\log(2/p)\}$.*

*Proof.* First, we note that if $X \sim \textbf{Lap}_{\mathbb{Z}}(1/\varepsilon)$, then it can rewritten as $X = N_1 - N_2$ where $N_1, N_2$ is geometrically distributed, i.e., $\mathbb{P}[N_1 = k] = \mathbb{P}[N_2 = k] = \beta^k(1 - \beta)$ with $\beta = e^{-\varepsilon}$. We can also write it as $X = \tilde{N}_1 - \tilde{N}_2$, where $\tilde{N}_1 = N_1 - \mathbb{E}[N_1]$ and $\tilde{N}_1 = N_2 - \mathbb{E}[N_2]$, since $\mathbb{E}[N_1] = \mathbb{E}[N_2]$. Then, by (Hillar & Wibisono, 2013, Lemma 4.3) and the equivalence of different definitions of sub-exponential (cf. Proposition 2.7.1 in Vershynin (2018)), we have $\tilde{N}_1, \tilde{N}_2$ are $(c_1^2 \frac{1}{\varepsilon^2}, c_1 \frac{1}{\varepsilon})$-sub-exponential for some absolute constant $c_1 > 0$. Now, since $X = \tilde{N}_1 - \tilde{N}_2$, by the weighted sum of zero-mean sub-exponential (cf. Corollary 4.2 in Zhang & Chen (2020)), we have $X$ is $(c^2 \frac{1}{\varepsilon^2}, c \frac{1}{\varepsilon})$-sub-exponential for some absolute constant $c > 0$. Thus, by Lemma 9, we have for any $p \in (0, 1]$, let $v \ge \max\{\frac{c}{\varepsilon}\sqrt{2n \log(2/p)}, 2\frac{c}{\varepsilon}\log(2/p)\}$, with probability at least $1 - p$, $|\sum_i X_i| \le v$. $\qquad\square$

**Theorem 6** (Local Pure-DP via SecAgg). *Let $\mathcal{P} = (\mathcal{R}, \mathcal{S}, \mathcal{A})$ be a protocol such that $\mathcal{R}$ given by Algorithm 4 with $\eta_i \sim \textbf{Lap}_{\mathbb{Z}}(g/\varepsilon)$, $\mathcal{S}$ is any SecAgg protocol and $\mathcal{A}$ is given by Algorithm 5. Fix any $\varepsilon \in (0, 1)$, for each $b \ge 1$, choose $n = l(b), g = \lceil \varepsilon\sqrt{n} \rceil, \tau = \lceil \frac{cg}{\varepsilon}\sqrt{2n \log(2/p)} \rceil$ ($c$ is the constant in Lemma 5), $p = 1/T$ and $m = ng + 2\tau + 1$. Then, Algorithm 1 instantiated with protocol $\mathcal{P}$ is able to achieve $(\varepsilon, 0)$-DP in the local model with expected regret given by*

$$\mathbb{E}[Reg(T)] = O\left(\sum_{a \in [K]: \Delta_a > 0} \frac{\log T}{\varepsilon^2 \Delta_a}\right).$$

*Moreover, the communication messages per user before $\mathcal{S}$ are $O(\log m)$ bits.*

*Proof.* Privacy: By the privacy guarantee of discrete Laplace and the sensitivity of $\widehat{x}_i$, we directly have the local pure DP for our algorithm.

Regret: The first step is to show that with high probabilit, for a large batch size $n = l(b) \ge 2 \log(2/p)$,

$$\left| \mathcal{A}(\widehat{y}) - \sum_i x_i \right| \le O\left( \frac{1}{\varepsilon}\sqrt{l(b) \log(1/p)} \right), \tag{13}$$

where $\widehat{y} := \widehat{y}_a(b)$.

We again can divide the LHS of equation 13 into

$$\left| \mathcal{A}(\widehat{y}) - \sum_i x_i \right| \leq \underbrace{\left| \mathcal{A}(\widehat{y}) - \frac{1}{g} \sum_i \widehat{x}_i \right|}_{\text{Term (i)}} + \underbrace{\left| \frac{1}{g} \sum_i \widehat{x}_i - \sum_i x_i \right|}_{\text{Term (ii)}}.$$

In particular, Term (ii) can be bounded by using the same method, i.e., for any $p \in (0, 1]$, with probability at least $1 - p$,

$$\text{Term (ii)} \leq O\left( \frac{1}{\varepsilon} \sqrt{\log(1/p)} \right).$$

To bound Term (i), the key is again to bound the total noise $\sum_i \eta_i$, i.e.,

$$\mathbb{P}\left[ \left| \sum_{i \in [n]} \eta_i \right| > \tau \right] \leq p.$$

To this end, by Lemma 5, when $n = l(b) \geq 2\log(2/p)$ and $\tau = \frac{cg}{\varepsilon}\sqrt{2n\log(2/p)}$, the above concentration holds. Then, following the same steps as before, we can conclude that Term$(i) \leq O(\frac{1}{\varepsilon}\sqrt{2l(b)\log(1/p)})$ when $l(b) \geq 2\log(2/p)$.

We then make a minor modification of the proof of Lemma 2. The idea is simple: we divide the batches into two cases – one is that $l(b) \geq 2\log(2T)$ (noting $p = 1/T$) and the other is that $l(b) \leq 2\log(2T)$. First, one can easily bound the total regret for the second case. In particular, during all the batches such that $l(b) \leq 2\log(2T)$, the total regret is bounded by $O((K-1)\log T)$. For the first case where $l(b)$ is large, by equation 13 and the same steps in the proof of Lemma 2, we can conclude that

$$l(\tilde{b}_a) \leq \max\left\{ \frac{c_1 \log(KT/p)}{\Delta_a^2}, \frac{c_2 \log(KT/p)}{\varepsilon^2 \Delta_a^2} \right\},$$

where $\tilde{b}_a$ is the last batch such that the suboptimal arm $a$ is still active. Thus, putting everything together and noting that $\Delta_a \leq 1$ and $\varepsilon \in (0, 1)$, we have

$$\mathbb{E}\left[\text{Reg}(T)\right] = O\left( \sum_{a \in [K]: \Delta_a > 0} \frac{\log T}{\varepsilon^2 \Delta_a} \right).$$

$\square$

## I    TIGHT PRIVACY ACCOUNTING FOR RETURNING USERS VIA RDP

In this section, we demonstrate the key advantages of obtaining RDP guarantees compared to approximate DP in the context of MAB. The key idea here is standard in privacy literature and we provide the details for completeness. The key message is that using the conversion from RDP to approximate DP allows us to save additional logarithmic factor in $\delta$ in the composition compared to using advanced composition theorem.

In the main paper, as in all the previous works on private bandit learning, we focus on the case where the users are unique, i.e., no returning users. However, if some user returns and participates in multiple batches, then the total privacy loss needs to resort to composition. In particular, let us consider the following returning situations.

**Assumption 2** (Returning Users). *Any user can participate in at most $B$ batches, but within each batch $b$, she only contributes once.*

Note that the total $B$ batches can even span multiple learning process, each of which consists of a $T$-round MAB online learning.

Let's first consider approximate DP in the distributed model, e.g., binomial noise in Tenenbaum et al. (2021) via secure shuffling. To guarantee that each user is $(\varepsilon, \delta)$-DP during the entire learning process, by advanced composition theorem (cf. Theorem 7), we need to guarantee that each batch $b \in [B]$, it is $(\varepsilon_i, \delta_i)$-DP, where $\varepsilon_i = \frac{\varepsilon}{2\sqrt{2B\log(1/\delta')}}$, $\delta_i = \frac{\delta}{2B}$ and $\delta' = \frac{\delta}{2}$. Thus, for each batch the

variance of the privacy noise is

$$\sigma_i^2 = O(\log(1/\delta_i)/\varepsilon_i^2) = O\left(\frac{B\log(1/\delta)\log(B/\delta)}{\varepsilon^2}\right). \tag{14}$$

Now, we turn to the case of RDP, (e.g., obtained via Skellam mechanism in Theorem 2 or via discrete Gaussian in Theorem 4). To gain the insight, we again consider the case that the scaling factor $s$ is large enough such that for each batch $b$, it is $(\alpha, \frac{\alpha\varepsilon^2}{2})$-RDP for all $\alpha$, i.e., it is approximately $\frac{\varepsilon^2}{2}$-CDP. Thus, $B$ composition of it yields that it is now $\frac{B\varepsilon^2}{2}$-CDP, and by the conversion lemma (cf. Lemma 10), it is $(\varepsilon', \delta)$-DP with $\varepsilon' = O(\varepsilon\sqrt{B\log(1/\delta)})$. Thus, in order to guarantee $(\varepsilon, \delta)$-DP in the distributed model, the variance of the privacy noise at each batch is

$$\sigma_i^2 = O\left(\frac{B\log(1/\delta)}{\varepsilon^2}\right). \tag{15}$$

Comparing equation 14 and equation 15, one can immediately see the gain of $\log(B/\delta)$ in the variance, which will translate to a gain of $O(\sqrt{\log(B/\delta)})$ in the regret bound.

**Remark 8.** *For a more accurate privacy accounting, a better way is to consider using numeric evaluation rather than the above loose bound.*

**Remark 9.** *If one is interested in pure DP, then by simple composition, the privacy loss scales linearly with respect to $B$ rather than $\sqrt{B}$.*

## J    MORE DETAILS ON SIMULATIONS

We numerically compare the performance of Algorithm 1 under pure-DP and RDP guarantees in the distributed model (named Dist-DP-SE and Dist-RDP-SE, respectively) with the DP-SE algorithm of Sajed & Sheffet (2019), which achieves pure-DP under the central model. We vary the privacy level as $\varepsilon \in \{0.1, 0.5, 1\}$, where a lower value of $\varepsilon$ indicates higher level of privacy.

In Figure 2, we consider the *easy* instance, i.e., where arm means are sampled uniformly in $[0.25, 0.75]$. In Figure 3, we consider the *hard* instance, i.e., where arm means are sampled uniformly in $[0.45, 0.55]$. The sampled rewards are Gaussian distributed with the given means and truncated to $[0, 1]$. We plot results for $K = 10$ arms.

We see that, for higher value of time horizon $T$, the time-average regret of Dist-DP-SE is order-wise same to that of DP-SE, i.e., we are able to achieve similar regret performance in the distributed trust model as that is achieved in the central trust model. As mentioned before, we observe a gap for small value of $T$, which is the price we pay for discrete privacy noise (i.e., additional data quantization error on the order of $O(\frac{1}{\varepsilon}\sqrt{\log(1/p)})$) and not requiring a trusted central server. Hence, if we lower the level of privacy (i.e., higher value of $\varepsilon$), this gap becomes smaller, which indicates an inherent trade-off between privacy and utility.

We also observe that if we relax the requirement of privacy from pure-DP to RDP, then we can achieve a considerable gain in regret performance; more so when privacy level is high (i.e., $\varepsilon$ is small). This gain depends on the scaling factor $s$ – the higher the scale, the higher the gain in regret.

In Figure 4, we compare regret achieved by our generic batch-based successive arm elimination algorithm (Algorithm 1) instantiated with different protocols $\mathcal{P}$ under different trust models and privacy guarantees: (i) central model with pure-DP (CDP-SE), (ii) local model with pure-DP (LDP-SE), (iii) Distributed model with pure-DP (Dist-DP-SE), Renyi-DP (Dist-RDP-SE) and Concentrated-DP (Dist-CDP-SE). First, consider the pure-DP algorithms. We observe that regret performance of CDP-SE and Dist-DP-SE is similar (with a much better regret than LDP-SE). Now, if we relax the pure-DP requirement, then we achieve better regret performance both for Dist-RDP-SE and Dist-CDP-SE. Furthermore, Dist-CDP-SE performs better in terms of regret than Dist-RDP-SE. This is due to the fact that under CDP, we use discrete Gaussian noise (which has sub-Gaussian tails) as opposed to the Skellam noise (which has sub-exponential tails) used under RDP.

In Figure 5, we show clear plots for our experiment on bandit instances generated from Microsoft Learning to Rank dataset MSLR-WEB10K (Qin & Liu, 2013).

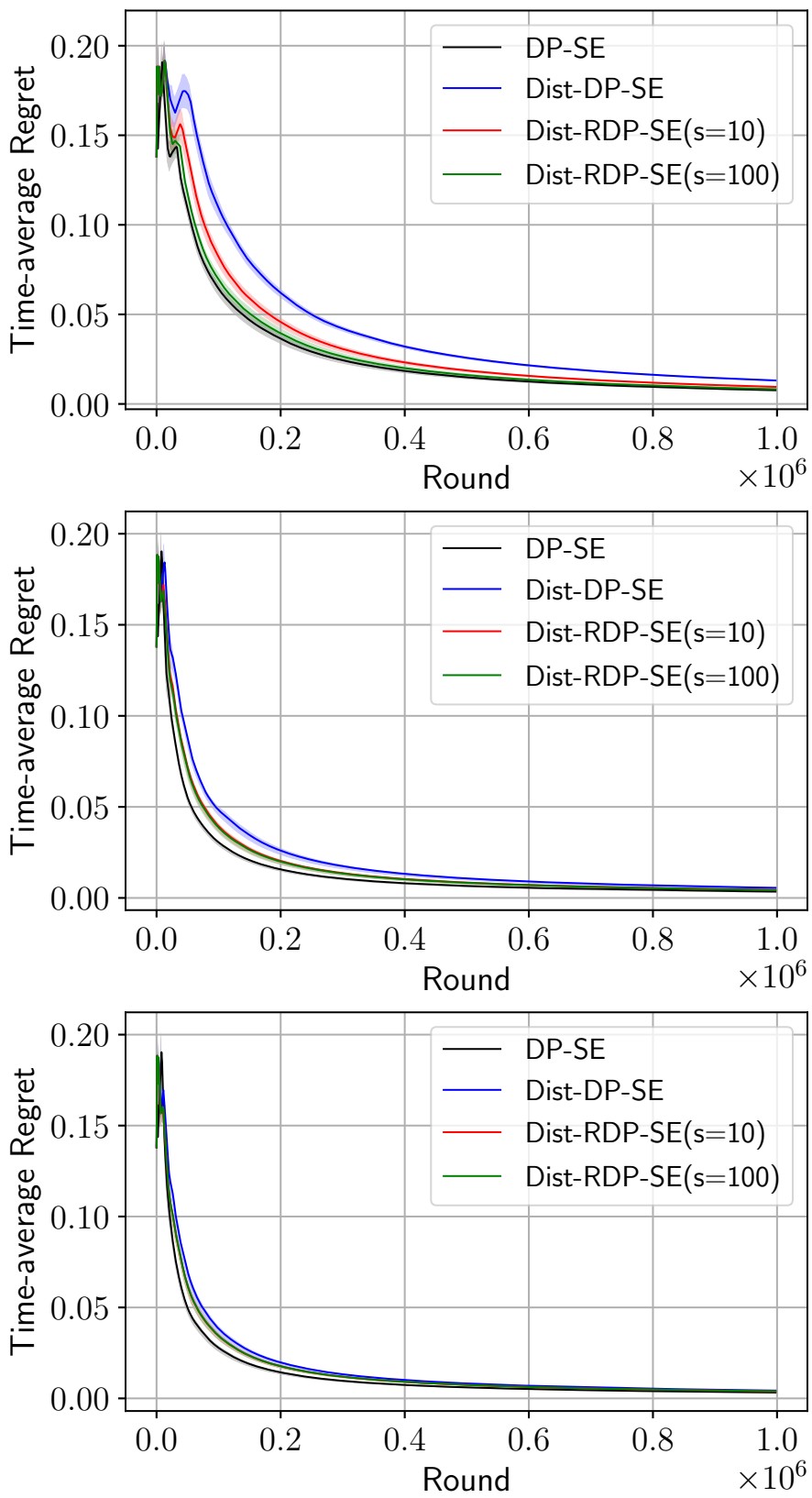

Figure 2: Comparison of time-average regret for Dist-DP-SE, Dist-RDP-SE, and DP-SE in Gaussian bandit instances under large reward gap (easy instance) with privacy level $\varepsilon = 0.1$ (top), $\varepsilon = 0.5$ (mid) and $\varepsilon = 1$ (bottom)

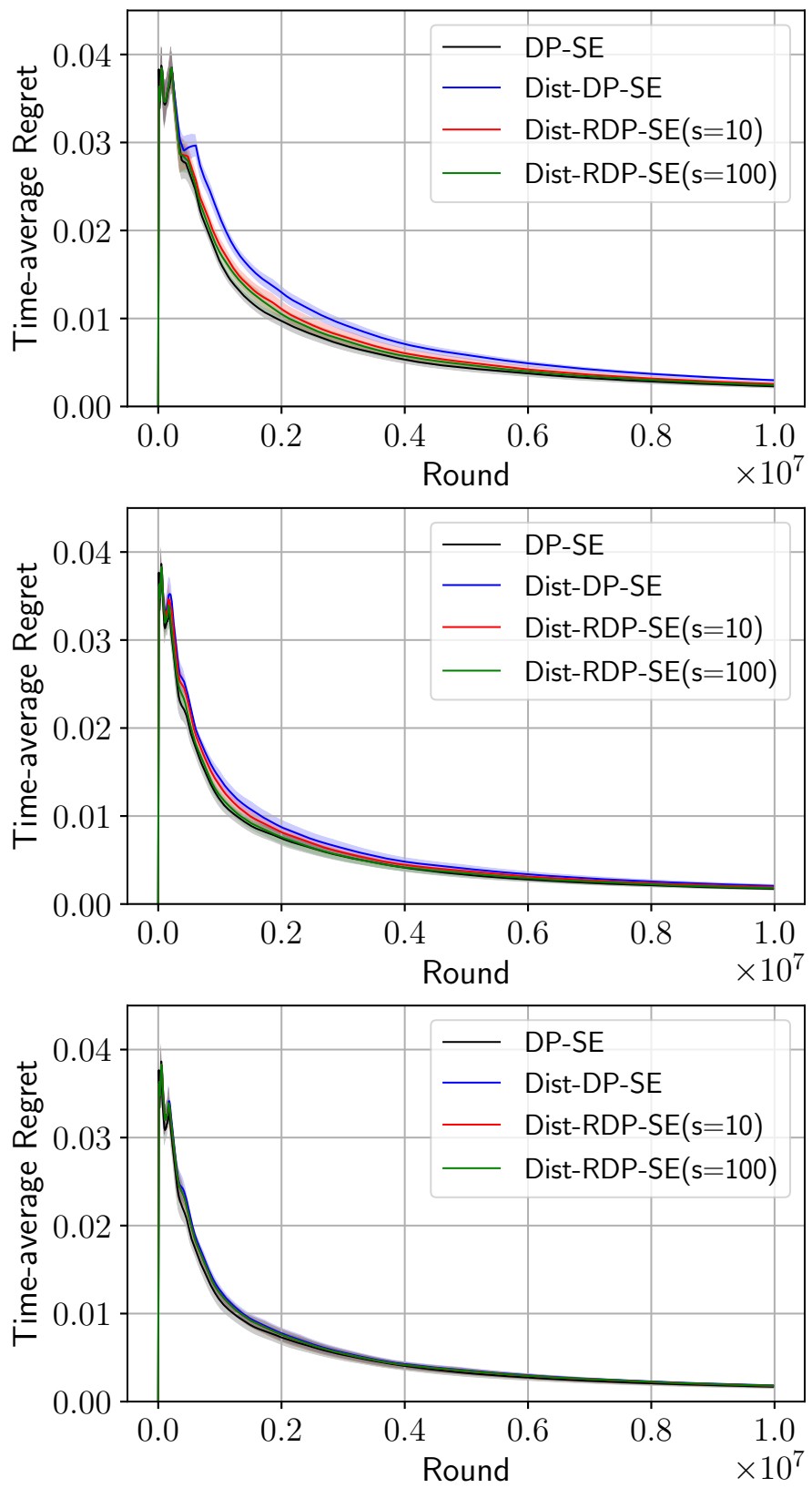

Figure 3: Comparison of time-average regret for Dist-DP-SE, Dist-RDP-SE, and DP-SE in Gaussian bandit instances under small reward gap (hard instance) with privacy level $\varepsilon = 0.1$ (top), $\varepsilon = 0.5$ (mid) and $\varepsilon = 1$ (bottom)

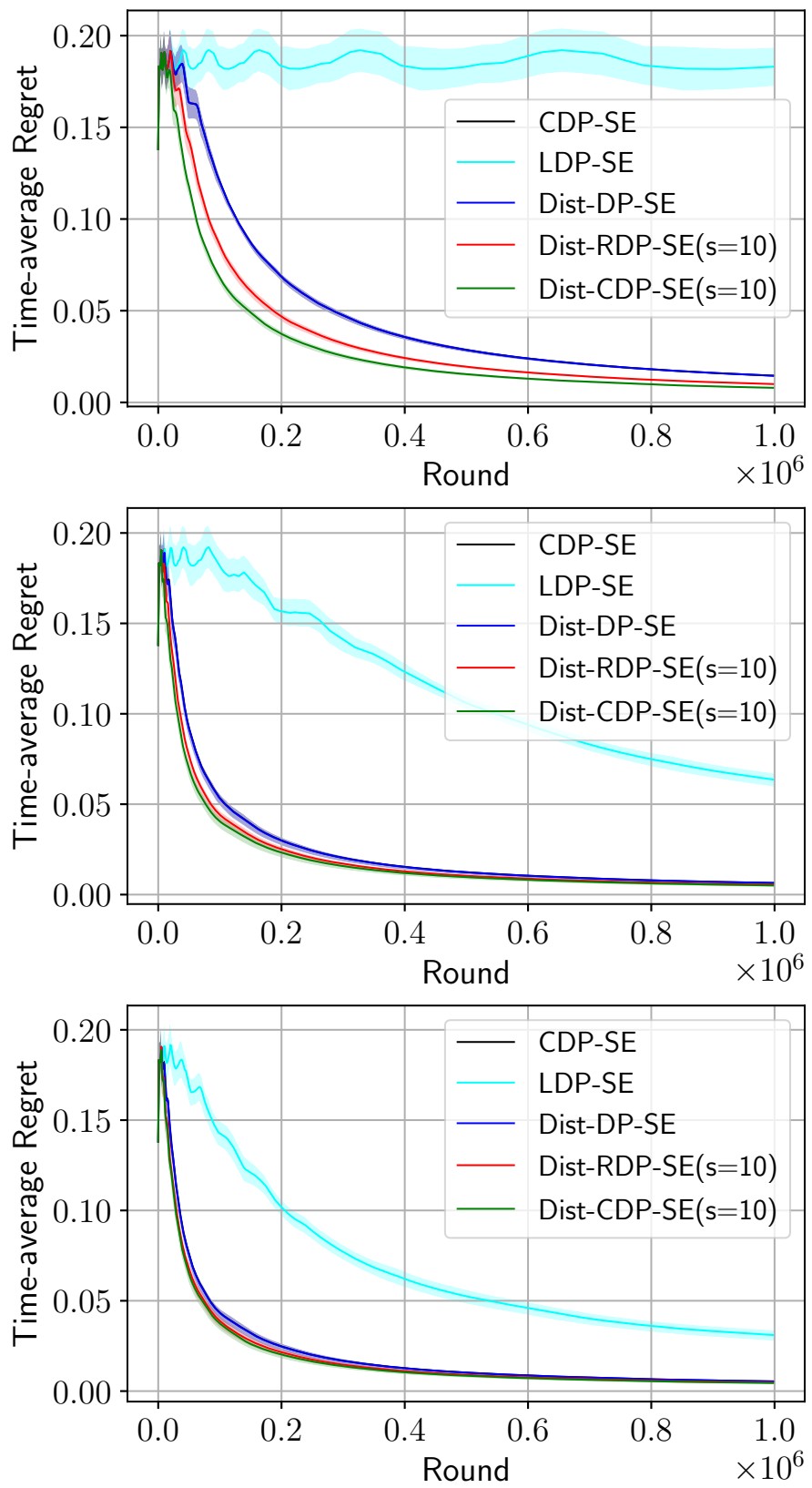

Figure 4: Comparison of time-average regret for CDP-SE, LDP-SE, Dist-DP-SE, Dist-RDP-SE and Dist-CDP-SE in Gaussian bandit instances under large reward gap (easy instance) with privacy level $\varepsilon = 0.1$ (top), $\varepsilon = 0.5$ (mid) and $\varepsilon = 1$ (bottom)

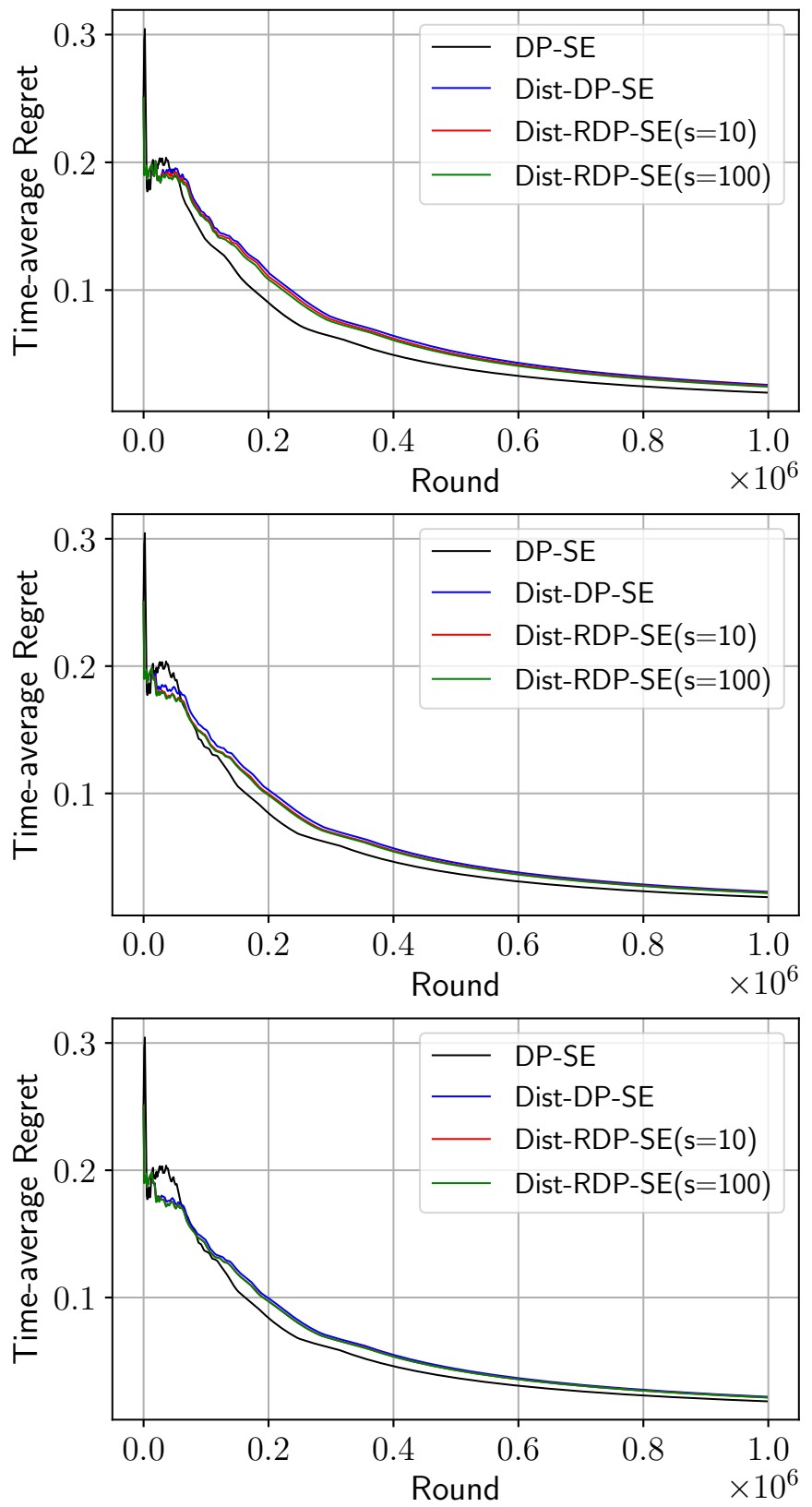

Figure 5: Comparison of time-average regret for Dist-DP-SE, Dist-RDP-SE, and DP-SE in bandit instances generated from real data with privacy level $\varepsilon = 1$ (top), $\varepsilon = 5$ (mid) and $\varepsilon = 10$ (bottom)

# K AUXILIARY LEMMAS

In this section, we summarize some useful facts that have been used in the paper.

**Fact 3** (Discrete Laplace Mechanism). *Let $\Delta, \varepsilon > 0$. Let $q : \mathcal{D}^n \to \mathbb{Z}$ satisfy $|q(D) - q(D')| \leq \Delta$ for all $D, D'$ differing on a single user's data. Define a mechansim $M : \mathcal{D}^n \to \mathbb{Z}$ by $M(D) = q(D) + Y$, where $Y \sim \textbf{Lap}_{\mathbb{Z}}(\Delta/\varepsilon)$. Then, $M$ satisfies $(\varepsilon, 0)$-DP. Moreover, for all $m \in \mathbb{N}$,*

$$\mathbb{P}\left[Y > m\right] = \mathbb{P}\left[Y < -m\right] = \frac{e^{-\frac{\varepsilon m}{\Delta}}}{e^{\frac{\varepsilon}{\Delta}} + 1}.$$

**Lemma 6** (Concentration of sub-Gaussian). *A mean-zero random variable $X$ is $\sigma^2$-sub-Gaussian if for all $\lambda \in \mathbb{R}$*

$$\mathbb{E}\left[e^{\lambda X}\right] \leq \exp\left(\frac{\lambda^2 \sigma^2}{2}\right).$$

*Then, it satisfies that for any $p \in (0, 1]$, with probability at least $1 - p$,*

$$|X| \leq \sqrt{2}\sigma\sqrt{\log(2/p)}.$$

**Lemma 7** (Concentration of sub-exponential). *A mean-zero random variable $X$ is $(\sigma^2, h)$-sub-exponential if for $|\lambda| \leq 1/h$*

$$\mathbb{E}\left[e^{\lambda X}\right] \leq \exp\left(\frac{\lambda^2 \sigma^2}{2}\right).$$

*Then, we have*

$$\mathbb{P}\left[|X| > t\right] \leq 2\exp\left(-\min\left\{\frac{t^2}{2\sigma^2}, \frac{t}{2h}\right\}\right).$$

*Thus, it satisfies that for any $p \in (0, 1]$, with probability at least $1 - p$,*

$$|X| \leq \sqrt{2}\sigma\sqrt{\log(2/p)} + 2h\log(2/p).$$

**Lemma 8** (Hoeffding's Inequality). *Let $X_1, \ldots, X_n$ be independent and identically distributed (i.i.d) random variables and $X_i \in [0, 1]$ with probability one. Then, for any $p \in (0, 1]$, with probability at least $1 - p$,*

$$\left|\frac{1}{n}\sum_{i=1}^{n} X_n - \mathbb{E}\left[X_1\right]\right| \leq \sqrt{\frac{\log(2/p)}{2n}}.$$

**Lemma 9** (Sum of sub-exponential). *Let $\{X_i\}_{i=1}^n$ be independent zero-mean $(\sigma_i^2, h_i)$-sub-exponential random variables. Then, $\sum_i X_i$ is $(\sum_i \sigma_i^2, h_*)$-sub-exponential, where $h_* := \max_i h_i$. Thus, we have*

$$\mathbb{P}\left[\left|\sum_i X_i\right| > t\right] \leq 2\exp\left(-\min\left\{\frac{t^2}{2\sum_i \sigma_i^2}, \frac{t}{2h_*}\right\}\right).$$

*In other words, for any $p \in (0, 1]$, if $v \geq \max\{\sqrt{2\sum_i \sigma_i^2 \log(2/p)}, 2h_* \log(2/p)\}$, with probability at least $1 - p$, $|\sum_i X_i| \leq v$.*

**Lemma 10** (Conversion Lemma). *If $\mathcal{M}$ satisfies $(\alpha, \varepsilon(\alpha))$-RDP, then for any $\delta \in (0, 1)$, $\mathcal{M}$ satisfies $(\varepsilon, \delta)$-DP where*

$$\varepsilon = \inf_{\alpha > 1} \varepsilon(\alpha) + \frac{\log(1/(\alpha\delta))}{\alpha - 1} + \log(1 - 1/\alpha).$$

*If $\mathcal{M}$ satisfies $\frac{1}{2}\varepsilon^2$-CDP, then for for any $\delta \in (0, 1)$, $\mathcal{M}$ satisfies $(\varepsilon', \delta)$-DP where*

$$\varepsilon' = \inf_{\alpha > 1} \frac{1}{2}\varepsilon^2 \alpha + \frac{\log(1/(\alpha\delta))}{\alpha - 1} + \log(1 - 1/\alpha) \leq \varepsilon \cdot \left(\sqrt{2\log(1/\delta)} + \varepsilon/2\right).$$

*Moreover, if $\mathcal{M}$ satisfies $(\varepsilon, 0)$-DP, then it satisfies $(\alpha, \frac{1}{2}\varepsilon^2\alpha)$-RDP simultaneously for all $\alpha \in (1, \infty)$.*

**Theorem 7** (Advanced composition). *Given target privacy parameters $\varepsilon' \in (0, 1)$ and $\delta' > 0$, to ensure $(\varepsilon', k\delta + \delta')$-DP for the composition of $k$ (adaptive) mechanisms, it suffices that each mechanism is $(\varepsilon, \delta)$-DP with $\varepsilon = \frac{\varepsilon'}{2\sqrt{2k\log(1/\delta')}}$.*

