# OpenReview forum: "Distributed Differential Privacy in Multi-Armed Bandits"
_ICLR.cc/2023/Conference — ICLR 2023 poster_

### Official Review · Reviewer_Z6zL · 2022-10-25

**Confidence:** 3
**Correctness:** 4
**Technical Novelty And Significance:** 3
**Empirical Novelty And Significance:** 3
**Recommendation:** 8

**Clarity, Quality, Novelty And Reproducibility:**

Regarding the clarity and the writing, I have the following points.
1. In the preliminaries for MAB, I would also want to describe the problem in the context of multiple users as in this paper. Doesn't really help if I have to look ahead in the paper for more context about this problem (even though this is quite well-known).
2. In Section 3 where you define $l(b)$. I think it's a confusing notation because $l(b)$ just gives us a number of users at time batch $b$? We might also want to incorporate arm $a$ in there somewhere. Does this mean that in a time batch $b$, $l(b)$ users will pull arm $a$ in the time duration of batch $b$? I'm unsure about this notation, to be honest. It shouldn't be the case that I have to read the algorithm to make sense of this. But then, you define it finally in the algorithm as $2^b$. So, does it mean that $T \geq K2^b$?
3. At the end of page 4, you pretty much start stating the three points that you did earlier on page 2. I don't think that's really necessary.

I like the quality of the work because the theoretical results produced are the best so far, and match the best possible utility. The empirical results look alright, too.

The novelty is what I'm concerned about, which I mention as Weakness 1 in the previous section. On one hand, it is nice to use existing and familiar techniques and analyses, but it does chip away from the significance sometimes, although I agree that it is a grey area to evaluate on.

**Strength And Weaknesses:**

Strengths:
1. The results for distributed DP provided here achieve the strongest privacy guarantees with utility matching that of the optimal, pure, central DP algorithm. The prior work in this distributed setting only offered approximate DP guarantees with utility worse by a multiplicative factor of $\sqrt{\log(1/\delta)}$. The idea of using Polya noise is quite nice!
2. The communication complexity of the above is also not too high -- $\log(m)$ bits.
3. The above could be extended to shuffle DP setting, as well.
4. The RDP guarantee is much better in terms of the utility, and the algorithm does well empirically on real-world datasets, too.
5. The above has asymptotically better utility than that of the prior work, has better guarantee than this paper's pure DP analogue for the setting where $\delta > 1/T$, and has better guarantees under composition than that of the prior work.
6. The experimental results indicate that these algorithms do almost as well as the central DP algorithm on both synthetic and real-world datasets.

Weaknesses:
1. The algorithmic framework is heavily inspired by those of the prior work of Balle et al' 2020 and Cheu et al' 2021. Even for the RDP algorithm, the idea of using Skellam noise is borrowed. So, I'm not sure how much the algorithmic novelty is here.
2. As acknowledged by the authors (which I appreciate), the communication cost gets worse in the RDP setting with better privacy and utility demands. Is that a necessity? Some discussion of that could be useful!
3. I have some complaints with the writing quality of this paper, which I will describe in the next section.

**Summary Of The Paper:**

This paper studies the multi-armed bandit (MAB) problem under the constraint of *distributed* differential privacy (DP). The goal here is to match the optimal utility guarantees under *central* DP, whilst being able to achieve stronger privacy guarantees in the model where we don't have a central, trustworthy server for aggregation and analysis. This paper overcomes the following limitations of prior work in this distributed setting (albeit via *shuffle* DP): (1) it only provides approximate DP guarantees; (2) the cost of privacy is a factor of $\sqrt{\log(1/\delta)}$ away from the optimal regret under pure DP in the central model; and (3) this works only for binary rewards or leads to high communication costs for real rewards.

There are two algorithms provided here: one for pure DP and one for Renyi DP (RDP). The four main challenges overcome and goals achieved here are: communication cost while achieving DP; achieving the best results so far under pure distributed DP; much better bounds under RDP, especially on real-world data; and overcoming the issue of privacy loss on finite computers due to floating point errors by using discrete noise distributions. Both theoretical and empirical results are provided.

**Summary Of The Review:**

I think it's a decent paper, with potential improvements out there in the writing. Based on what I've said so far, I feel accepting it should be alright, but I would be taking the authors' response into account before sending my final recommendation.

Edit: updated my score now.

---

> ### Author Response · Authors · 2022-11-14
> **Response to Reviewer Z6zL**
>
> Thanks for taking the time to review our work and provide valuable comments. We are glad to hear that you like the quality of our work and think it is nice to use existing and familiar techniques to establish new results. We will recap your comments and present our detailed response.  We hope our answers will resolve your concern.
>
> ---
> > The algorithmic framework is heavily inspired by those of the prior work of Balle et al' 2020 and Cheu et al' 2021. Even for the RDP algorithm, the idea of using Skellam noise is borrowed. So, I'm not sure how much the algorithmic novelty is here.
>
> We agree with the reviewer that the privacy block of the algorithm is inspired by prior work. Our main novelty lies in integrating this into the learning block of the algorithm. Note that prior work focuses on supervised learning, while we consider a bandit learning problem, which presents different challenges. As already highlighted in the paper, regret analysis in bandits requires a (tight) tail bound compared to supervised learning where a bound on the noise variance is sufficient to analyze utility. For example, we need to prove a new tail bound for Skellam noise to derive a regret guarantee under RDP.
> Furthermore, additional operations (fixed-point encoding, randomized rounding, modular clipping) in the distributed DP model make the tail-bound analysis more delicate compared to the central model.
> Finally, as a by-product of our main results, we also recover the optimal regret bounds under central and local models while only using discrete privacy noise. This helps to avoid potential privacy leakage of continuous privacy noise on finite computers in practice.
>
>
> ---
> > As acknowledged by the authors (which I appreciate), the communication cost gets worse in the RDP setting with better privacy and utility demands. Is that a necessity? Some discussion of that could be useful!
>
> We first recall that the communication cost for RDP with SecAgg is roughly  $O(\log(n+ s/\epsilon))$, where $n$ is the batch size and $s$ is the scaling factor. A large $s$ leads to better privacy and regret as shown in Theorem 2, but incurs a larger communication.
>
> However, we tend to believe that one can break the privacy-communication trade-off above using a very recent technique proposed in [R1]. In particular, [R1] shows that the fundamental communication cost for RDP with SecAgg for the mean estimation task under mean-square error utility metric scales with $\Omega(\max(\log(n^2\epsilon^2), 1))$, where $n$ is the batch size. That is, for a stronger privacy guarantee (i.e., a smaller $\epsilon$), each user should send less number of bits. The intuition is that if a user sends less bits, then she communicates less information about her local data (hence protection of privacy). To achieve this improvement, [R1] proposes to use a linear compression scheme based on sparse random projections and distributed discrete Gaussian noise. Now, to apply the same technique to our private bandit problem, one needs to handle a different utility metric -- that is, instead of the mean-square error in [R1], one now needs to examine the tail concentration behavior. We leave it as an interesting future work.
>
> [R1] Chen, Wei-Ning, Christopher A. Choquette Choo, Peter Kairouz, and Ananda Theertha Suresh. "The fundamental price of secure aggregation in differentially private federated learning." ICML, 2022.
>
>
> ---
> > 1. In the preliminaries for MAB, I would also want to describe the problem in the context of multiple users as in this paper. Doesn't really help if I have to look ahead in the paper for more context about this problem (even though this is quite well-known).
>
> Thanks for the suggestion. We will incorporate this in the final version. The crux here is that each round $t$ is associated with a unique user.
>
>
> ---
> > 2. In Section 3 where you define $l(b)$. I think it's a confusing notation because just gives us a number of users at time batch $b$? We might also want to incorporate arm $a$ in there somewhere. Does this mean that in a time batch $b$,  users will pull arm $a$ in the time duration of batch $b$? I'm unsure about this notation, to be honest. It shouldn't be the case that I have to read the algorithm to make sense of this. But then, you define it finally in the algorithm as $2^b$. So, does it mean that $T \ge K 2^b$?
>
> We understand your concern. Indeed, one can introduce $l_a(b)$ for each active arm $a \in \Phi(b)$. In the current notation, we have $l_a(b)=l(b)=2^b$ for each $a \in \phi(b)$. Hence, we mentioned that for each active arm $a$, the algorithm recommends it to a **new** batch of users with the size being $l(b)$, see Line 6 in our algorithm. To avoid confusion, we will make the change in the final version.
>
>
> ---
> > 3. At the end of page 4, you pretty much start stating the three points that you did earlier on page 2. I don't think that's really necessary.
>
> Thanks, we will remove them in the final version.

---

### Official Review · Reviewer_PLdo · 2022-10-25

**Confidence:** 3
**Correctness:** 4
**Technical Novelty And Significance:** 3
**Empirical Novelty And Significance:** 3
**Recommendation:** 6

**Clarity, Quality, Novelty And Reproducibility:**

The paper is clearly written. The results are new, and the algorithmic techniques are practically relevant.

**Strength And Weaknesses:**

Strength
- This work fills in the gap for distributed pure DP in previous works and the result is tight (matches the central DP lower bound)
- The RDP upper bound is promising, and the result is relevant in practice as RDP is often used for tight privacy accounting.
- The algorithm uses discrete noise which is more relevant in practice.
- The theoretical bounds are verified by experiment results.

Weakness
- I was wondering if the distributed DP bounds can be obtained from LDP bound + privacy amplification by shuffling.
- Lower bound is not provided for RDP.
- Technically the work seems to mainly use existing techniques (discrete Laplace, Skellam noise, etc.). Perhaps the authors could better explain the technical novelties, e.g. why $(\varepsilon, 0)$ bound could not be obtained in previous works and how it is obtained in this work


Minor comments:
Since the upper bound for distributed $(\varepsilon, 0)-$DP matches that of central DP, row 4 in Table 1 could be changed to $\Theta$ instead of $O$

**Summary Of The Paper:**

This work proves regret bounds for multi-arm bandit under distributed pure DP and RDP only using discrete noise. The theoretical results are verified by experiments.

**Summary Of The Review:**

This work provides new regret bounds in MAB under distributed DP verified by experiments. The main drawback is that lower bound is missing for RDP, and technical novelty is unclear.

---

> ### Author Response · Authors · 2022-11-14
> **Response to Reviewer PLdo**
>
> Thanks for taking the time to review our work and provide valuable comments. We are glad to hear that you find our algorithmic techniques practically relevant. We will recap your comments and present our detailed response.  We hope our answers will resolve your concern.
>
> ---
> > I was wondering if the distributed DP bounds can be obtained from LDP bound + privacy amplification by shuffling.
>
> To the best of our understanding, the state-of-the-art *general* LDP amplification by shuffling (see [R1]) cannot give pure DP. Since our main focus is to achieve pure DP, we leverage recent advances in secure aggregation and a *specific* secure shuffling protocol [CY'21].
>
>
> [R1] Feldman, Vitaly, Audra McMillan, and Kunal Talwar. “Stronger Privacy Amplification by Shuffling for Renyi and Approximate Differential Privacy.” arXiv preprint arXiv:2208.04591 (2022).
>
>
>
> ---
> > Lower bound is not provided for RDP.
>
> We first note that even in the central model, private MAB with RDP has not been explicitly studied before and the regret lower bound is still not known. One promising approach to establish the lower bound in the central model for RDP could be deriving a notion of *private* KL-divergence under RDP by adapting the one for pure DP in [R1]. Then, this established lower bound under the central model is also a valid one for RDP under the distributed model.
> We leave this as an interesting future work.
>
> [R1] Azize, Achraf, and Debabrota Basu. "When Privacy Meets Partial Information: A Refined Analysis of Differentially Private Bandits." arXiv preprint arXiv:2209.02570 (2022).
>
>
> ---
> > Technically the work seems to mainly use existing techniques (discrete Laplace, Skellam noise, etc.). Perhaps the authors could better explain the technical novelties, e.g. why $(\epsilon,0)$ bound could not be obtained in previous works and how it is obtained in this work
>
> We first explain why $(\epsilon,0)$ privacy guarantee is not achieved in previous work in the distributed model. In previous work [TKMS'21], the authors use private summation protocol via shuffling, which builds upon and amplifies the binomial mechanism. The binomial mechanism only gives $(\epsilon,\delta)$-DP. Furthermore, even if one replaces binomial noise with Polya noise, a straightforward application of shuffling again can only give approximate DP (as shown in [BBGN'20]). To achieve pure DP in the distributed model, we adopt two new approaches. One is a new secure protocol -- SecAgg, which enables us to obtain discrete Laplace from the sum of Polya noise, and hence obtain pure DP. Another one is via a new shuffling protocol, which aims to simulate a *relaxed* SecAgg, hence again enabling us to achieve pure DP.
>
> We then clarify that to achieve RDP via Skellam noise, the existing technique is not sufficient for us to establish the regret bound. In particular, in the supervised learning setting of prior work, a bound on the second moment is sufficient to show a utility guarantee. In contrast, in our MAB setting, we need a bound on the tail of the Skellam random variable, which, to the best of our knowledge, is not shown in prior work. Thus, we first show that Skellam has a sub-exponential tail, which not only enables us to obtain the regret bound under RDP, but also could be of independent interest.
>
> We will add the above discussion in the final version to better distinguish our work from prior work.
>
> [BBGN’20] Borja Balle, James Bell, Adria Gascon, and Kobbi Nissim. Private summation in the multi-message shuffle model. In Proceedings of the 2020 ACM SIGSAC Conference on Computer and Communications Security, pp. 657–676, 2020.
>
> ---
> > Minor comments:  Since the upper bound for distributed DP matches that of central DP, row 4 in Table 1 could be changed to $\Theta$  instead of $O$.
>
> Thanks for the suggestion. We will update it in the final version.

---

> > ### Comment · Reviewer_PLdo · 2022-11-21
> > **Thanks for your response**
> >
> > Thanks very much for your response. My score remains unchanged.

---

### Official Review · Reviewer_V7G1 · 2022-10-31

**Confidence:** 3
**Correctness:** 4
**Technical Novelty And Significance:** 3
**Empirical Novelty And Significance:** Not applicable
**Recommendation:** 8

**Clarity, Quality, Novelty And Reproducibility:**

The initial submission did not clearly explain the innovation in this paper over prior work. The latest revision, however, significantly improves in this aspect.

**Strength And Weaknesses:**

Getting a tight bound for a fundamental problem, in a distributed model of differential and yet matching the central model lower bound, is certainly nice.

The paper builds on ideas from prior works. The main algorithm is very similar but different from the VB-SDP-AE algorithm from the Tenebaum, Kaplan, Mansour, Stemmer [TMMS] paper. Part of the improvement over [TMMS] comes from using a better primitive for private binary summation, either by using a different secure intermediary primitive (secure aggregation) or by utilizing a more recent private summation protocol for the shuffle model, from a preprint of Cheu and Yan. Nevertheless, this is not enough to get the tight bound, and the additional tweak added to the TMMS algorithm seems to be essential.

**Summary Of The Paper:**

This paper considers the (stochastic) multiarmed bandits problems in the setting of distributed differential privacy, where each individual’s reward function needs to be protected, and the central server running the bandits algorithm is not fully trusted to receive private information in the clear.  The setting studied in the paper does, however, allow for a secure intermediary, such as secure aggregation, or shuffling.

The main result of the paper is an algorithm with excess regret of $O(\varepsilon^{-1} k \log T)$, where $k$ is the number of arms, $T$ is the number of time steps, and $\varepsilon$ is the privacy parameter. This is optimal for pure differential privacy even in the central model, and effectively removes a $\sqrt{log 1/\delta}$ factor from prior work. The prior work also offered somewhat weaker privacy guarantees.

**Summary Of The Review:**

The paper has a nice result, which builds on prior work, but also adds new ideas.

---

> ### Author Response · Authors · 2022-11-14
> **Response to Reviewer V7G1**
>
> Thanks for taking the time to review our work and provide valuable comments. We are glad to hear that you find our results on matching the central model regret to be nice. We will recap your comments and present our detailed response.  We hope our answers will resolve your concern.
>
> ---
> > The reason my rating is not higher is that I am not sure where the improvement comes from and whether it is really due to this paper. The main algorithm is essentially equivalent to the VB-SDP-AE algorithm from the Tenebaum, Kaplan, Mansour, Stemmer paper. As far as I understand, VB-SDP-AE paper is in the weaker approximate differential privacy model, and has the extra $\sqrt{\log (1/\delta)}$ factor in the excess regret because these were the best guarantees available at the time for the problem of private binary summation in the shuffle model of differential privacy. The present paper seems to circumvent this bottleneck by either using a different secure intermediary primitive (secure aggregation) or by utilizing a more recent private summation protocol for the shuffle model, from a preprint of Cheu and Yan. So doesn’t the result claimed in the paper follow more or less directly from the Tenenbaum et al. algorithm and the Cheu and Yan result?
>
> There is one key difference between our algorithm and the VB-SDP-AE algorithm in [TKMS'21]. At the start of a batch, VB-SDP-AE uses all the past data to compute reward estimates. In contrast, we adopt the idea of **forgetting** and use only the data of the last completed batch to design the estimates. This small but important tweak allows us to accumulate “less” noise in reward estimates compared to VB-SDP-AE, which is a key step to achieving optimal regret scaling. There are two immediate implications of this tweak.
>
> - If one directly applies the same shuffle protocol in [TKMS'21] to our algorithm, one can trim down the privacy cost from $O\left(\frac{K\log T\sqrt{\log(1/\delta)}}{\epsilon}\right)$ (as reported in [TKMS'21]) to $O\left(\frac{K\sqrt{\log T\log(1/\delta)}}{\epsilon}\right)$ in the weaker
> approximate DP model. Since our main focus in this paper is to achieve optimal regret in the pure DP model, we use secure aggregation and a specific secure shuffling protocol.
>
> - If one adapts the secure protocols for pure DP used in our paper to the VB-SDP-AE algorithm in [TKMS’21], one **cannot** achieve the optimal regret bound under pure DP due to the accumulation of privacy noise.
>
> Note that the effect of this tweak is also visible in our RDP to Approximate DP conversion guarantee -- we obtain the desirable $O\left(\frac{K\sqrt{\log T\log(1/\delta)}}{\epsilon}\right)$ privacy cost by virtue of Theorem 2 (see page 7).
>
> We will add the above discussion in the final version to better distinguish our result from prior work.
>
> ---
> > The paper has a nice result, but it’s unclear if that result follows directly from combining prior works, or if it required new ideas, and how these ideas differ from what was done before.
>
> As discussed above, a direct combination of prior work will not yield the optimal regret bound. Moreover, as stated in our introduction, to apply the recent advances in secure protocols to our bandit learning, one needs to address additional challenges of establishing a tight tail bound rather than a bound on second moment in supervised learning. For example, to establish the first regret bound under RDP with Skellam noise, we need to prove a novel tail bound of the Skellam random variable, which, to the best of our knowledge, is not shown in prior work. In addition to new theoretical results, our results also have meaningful practical importance. For example, as a by-product of our main results, we also recover the optimal regret bounds under central and local models while only using discrete privacy noise. This helps to avoid potential privacy leakage of continuous privacy noise on finite computers in practice.

---

> > ### Comment · Reviewer_V7G1 · 2022-11-14
> > **Thank you for clarifying the relationship to prior work**
> >
> > Thank you for these clarifications! I think it would help the reader if you do incorporate them into the paper.

---

> > > ### Author Response · Authors · 2022-11-15
> > > **Thank you for your feedback**
> > >
> > > We would like to thank you again for the insightful and constructive comment, which helps us to better distinguish our work from prior work (please see our updated version). We just wanted to check whether our new version and response have resolved your concern and helped you feel more positive about our paper. If you feel that your concerns have been answered, we would appreciate it if you could raise the score.

---

### Author Response · Authors · 2022-11-15
**A new version of the paper is uploaded**

Dear Reviewers and Area Chairs,

We realized it is allowed to update the paper during the rebuttal process. Hence, we have updated it (and the full paper in supplementary) by incorporating the valuable comments and suggestions of the reviewers. In particular, we mark the new changes using red color.

We hope this new version will help to (i) highlight the key difference compared to prior work and (ii) improve the presentation.

We are always open to making additional updates to the paper if the reviewers have further suggestions on the presentation.


Best,

Authors

---

### Decision · Program_Chairs · 2023-01-20

**Decision:**

Accept: poster

**Justification For Why Not Higher Score:**

Not a very wide audience for this paper.

**Justification For Why Not Lower Score:**

I think the combination of these areas, as well as the results, are interesting.


**Metareview: Summary, Strengths And Weaknesses:**

This paper has a clear contribution in a natural intersection between differential privacy and bandits: the multi-agent bandit setting. Everybody argues for acceptance.


**Note From Pc:**

if the above contains the word "oral" or "spotlight" please see: "oral" presentation means -> notable-top-5% and "spotlight" means -> notable-top-25%. As stated in our emails, we are disassociating presentation type from AC recommendations